# Flywheel eccentric overload exercises versus barbell half squats for basketball players: Which is better for induction of post-activation performance enhancement?

**Hezhi Xie[1‡], Wenfeng Zhang[2‡], Xing Chen[2], Jiaxin He[2], Junbing Lu[2], Yuhua Gao[1], Duanying Li[1], Guoxing Li[1], Hongshen Ji[1], Jian Sun**  [1]*

**1** School of Athletic Training, Guangzhou Sport University, Guangzhou, P.R. China, **2** Graduate School, Guangzhou Sport University, Guangzhou, P.R. China

‡ HX and WZ are co-first authors on this work.
* sunjian@gzsport.edu.cn

**Data Availability Statement:** All relevant data are within the manuscript and its Supporting Information files.

## Abstract

### Objective

This study compared the post-activation performance enhancement (PAPE) effects of a flywheel eccentric overload (FEOL) exercise and barbell half squats (BHS) on countermovement jump (CMJ) and 30 m sprint performance.

### Methods

Twelve male collegiate competitive basketball players were enrolled in this study and they implemented two training protocols: barbell half squat (BHS) and flywheel eccentric overload (FEOL) training. The BHS protocol included three intensities of load: low (40% 1RM), medium (60% 1RM), and high (80% 1RM), with each intensity consisting of 5 sets of 3 repetitions. The FEOL protocol included three inertia intensities: low (0. 015 kg·m²), medium (0.035 kg·m²), and high (0.075 kg·m²), with each intensity consisting of 3 sets of 6 repetitions. The measurement time points were before training (baseline) and at 3, 6, 9, and 12 minutes after training. A two-stage (stage-I and stage-II) randomized crossover design was used to determine the acute effects of both protocols on CMJ and sprint performance.

### Results

At each training intensity, the jump height, jump peak power output (PPO), jump impulse and 30m sprint speed at 3, 6, 9, and 12 minutes after BHS and FEOL training did not change significantly compared to the baseline. A 2-way ANOVA analysis indicated significant main effects of rest intervals on jump height, jump PPO, and jump impulse, as well as 30m sprint speed. The interaction of the Time × protocol showed a significant effect on jump height between BHS and FEOL groups at high intensity in stage-I (F = 3.809, p = 0.016, df = 4) and stage-II (F = 3.044, p = 0.037, df = 4). And in high training intensity, the jump height at 3 (7.78 ± 9.90% increase, ES = 0.561), 6 (8.96 ± 12.15% increase, ES = 0.579), and 9 min

**Funding:** The authors received no specific funding for this work.

**Competing interests:** The authors have declared that no competing interests exist.

(8.78 ± 11.23% increase, ES = 0.608) were enhanced in I-FEOL group compared with I-BHS group (F = 3.044, p = 0.037, df = 4). In stage-II, the impulse and sprint speed of the FEOL group were significantly higher than those of the BHS group at 6, 9, and 12 min under low (FEOL = 0.015kg·m$^2$, BHS = 40%1RM), medium(FEOL = 0.035kg·m$^2$, BHS = 60% 1RM), and high (FEOL = 0.075kg·m$^2$, BHS = 80%1RM) intensities. Furthermore, the sprint speed of the two training protocols did not change at different time points. The interaction of Time × training intensity showed lower sprint speeds in the II-BHS group at a high intensity (BHS = 80%1RM) compared to low (BHS = 40%1RM) and medium (BHS = 60%1RM) training intensities, especially at 9 min and 12 min rest intervals.

## Conclusion

Although barbell half squat training and flywheel eccentric overload training did not provide a significant PAPE effect on explosive power (CMJ and sprint) in male basketball players, FEOL training showed a better potential effect on enhanced CMJ jump performance at the high training intensity.

## Introduction

The characteristics of basketball require players' high levels of neuromuscular abilities including power output, strength, and speed [1]. Both strength and power training programs for basketball players should focus on developing the performance of explosive movements [2]. An acute enhancement in explosive sport performance namely throws, sprints and jumps can be achieved due to the muscular phenomenon called post-activation potentiation (PAP) [3] and an alternative term has been proposed in 2017, a post-activation performance enhancement (PAPE) [4]. Despite both are helpful in the increment of sports performance, differences exist in underlying physiological mechanisms and in the time of enhancement persistence. The primary underlying PAP mechanism is the phosphorylation of the myosin regulatory light chain and a very short period (<3 min) [5–8], while PAPE would be associated with other potential mechanisms namely muscle temperature [9], water content [10] and a longer "window of opportunity" (>4 min) [4, 11] as well as the presence of possible neural mechanisms [12, 13]. In training practice, the potentiation can be achieved by the execution of biomechanically similar conditioning activity (CA) [14] at maximal or near-maximal intensities prior to the subsequent athletic task. As our further in-depth understanding of the physiological mechanism of acute effects in muscle contractile function, many studies have indicated that an appropriate induction protocol can successfully induce PAPE.

Squat exercises can induce PAPE in the lower limbs. Most representative studies have reported positive effects of weighted squat training on athletes' performance in jump height, rate of force development, response time, maximal power, and maximal speed [15–18]. High and moderate weighted squats (90% and 60% of 1RM, respectively) have been reported to improve sprint and countermovement jump (CMJ) performance in male subjects [19, 20]. Additionally, flywheel ergometer squat training can also enhance the acute effects of strength, power, sprint, jumps, and maximal voluntary contraction [21–27]. These play an important role in most of the necessary movements in the sport.

Flywheel ergometer squat training stimulates eccentric overload training (EOL), where the eccentric muscle force generated (e.g., flexion phase of the squat) exceeds the maximum

concentric force (e.g., extension phase of the squat) [28–30]. In conventional squat training, there is a short pause between the ECC and CON movements. In contrast, the movements in the flywheel squat pattern are coupled and involve a stretch-shortening cycle, which may facilitate CON muscle activation and strength building [31]. However, eccentric overload has been reported to appear to be a means of increasing eccentric power and rate of force development (RFD) rather than concentric output [32]. Currently, there is only 1 literature comparing the acute effects of PAPE on physical performance induced by conventional squats and flywheel ergometers respectively, and only one training intensity [33]. Therefore, there is a need for a longitudinal study of both training protocols (with multiple intensities).

Previous studies on the PAPE effect have mostly focused on explosive power-based sports such as throwing [34], weightlifting [35], soccer [36], rugby [15], and track and field [37]. The effects of PAPE activation induced by barbell squatting and EOL training in elite basketball players over a season are rarely reported. This study aimed to compare the influence of exercise modality (FEOL or BHS) and intensity (low (FEOL = 0.015kg·m$^2$, BHS = 40%1RM), moderate (FEOL = 0.035kg·m$^2$, BHS = 60%1RM), and high (FEOL = 0.075kg·m$^2$, BHS = 80%1RM)) on PAPE in CMJ and sprint in basketball players.

## Methods

### Experimental approach to the problem

A two-phase randomized crossover study was used. Participants underwent a total of 12 training sessions lasting 13 weeks, with a washout period at week 7. The subjects had one additional familiarization session prior to the experiment. In stage-I (weeks 1–6), subjects were randomized into two groups (I-FEOL vs. I-BHS). In stage-II (weeks 8–13), the training modalities of the two groups of subjects were swapped, that is, I-FEOL was replaced by II-BHS, and I-BHS was replaced by II-FEOL. To compare the acute effects of the FEOL protocol and BHS protocol on CMJ and sprint performance after each training session (each intensity), the subjects underwent two rounds of either the FEOL protocol or the BHS protocol in each stage. The measurement time points for CMJ or sprint performance were set at baseline and at 3, 6, 9, and 12 minutes after each training session. We assumed that the low, medium or high resistance in the FEOL and BHS protocols were equivalent. The results are therefore inherently based on how well the intensities are matched. Each protocol contained three intensities: low (FEOL = 0.015kg·m$^2$, BHS = 40%1RM), medium (FEOL = 0.035kg·m$^2$, BHS = 60%1RM), and high (FEOL = 0.075kg·m$^2$, BHS = 80%1RM), with one intensity implemented each week. First round (1–3 weeks) of stage-I included week 1 = low, week 2 = medium, week 3 = high, and jump performance was measured. Second round (weeks 4–6) of stage-I included week 4 = low, week 5 = medium, week 6 = high, and sprint performance was measured. Phase two crossover trials. Round one (8–10 weeks) of stage-II was week 8 = low, week 9 = medium, week 10 = high, and jump performance was measured. Second round (weeks 11–13) of stage-II was week 11 = low, week 12 = medium, week 13 = high, and sprint performance was measured (shown in **Fig 1**).

The authors interrupted one training session (week 7) as a washout period (14 days between training sessions) to mitigate the effects of the first program on the second one, which were considered that the duration of the washout period needed further discussion. Participants did not alternate between CMJ and sprint tests during the first 3 training sessions based on the hypothesis that cross-validation after one training session (particularly in the FEOL protocol) might induce some muscle impairments that could lead to biased maximal performance at CMJ or sprint.

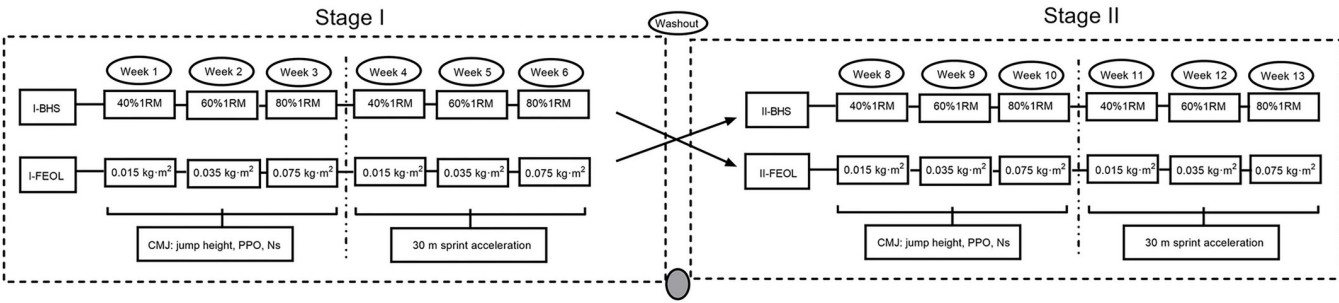

**Fig 1. Flowchart of the experiment.**

## Participants

Eighteen players (all team members) of the basketball team from Guangzhou Sport University, participating in the National University Basketball League (CUBA), volunteered to participate in this test. The inclusion criteria were as follows: the absence of any injury or disease [38] (Physical Activity Readiness Questionnaire), regular participation in basketball team training (4 training sessions of 120 min each per week), 3 basketball training sessions (Monday, Wednesday, and Saturday), 1 physical training session (Friday), more than 1 year of heavy strength resistance training experience, and assessed by 2 physical training specialists (mean experience of 12 years) for their fitness status. The players in this trial typically performed resistance training in the strength and conditioning laboratory once a week as a training task, and therefore had a priori knowledge of the training protocol and testing methods. All subjects were informed about the potential risks and benefits of the current procedures and signed an informed consent form. The players were also asked to maintain habitual exercise for 48 h before the trial and not to intake any stimulant substances or alcohol. Finally, twelve participants (age 20.63 ± 1.32 years, range 19–23 years; body mass 21.96 ± 1.36 kg; height 187.56 ± 5.55 cm) were included in the final analysis (Table 1). This study was approved by the Academic Ethics Committee of Guangzhou Sport University (2021LCLL-11). All procedures were conducted according to the Declaration of Helsinki for studies involving human participants.

## Procedures

During the first visit to the laboratory on a Friday afternoon one week prior to the trial, the basic information of all players was recorded. Height were measured using JENIX, DS-103M (Korea). Body and weight composition was recorded by experienced investigators using the InBody 370 body composition analyzer (Korea). Subsequently, the 1RM of a weighted half squat (knee angle 90˚) was measured for all subjects under the guidance of a physical training

**Table 1. Basic information of the participates.**

| Metrics | I-BHS (n = 6) | I-FEOL (n = 6) | Total | P |
|---|---|---|---|---|
| Age (years) | 20±1.41 | 21.25±1.26 | 20.63±1.32 | 0.235 |
| Height (cm) | 190.5±3.7 | 184.63±3.15 | 187.06±3.55 | 0.161 |
| Body weight (kg) | 78.4±4.14 | 66.73±2.26 | 72.56±6.51 | 0.003** |
| Body Mass Index(kg/m$^2$) | 23.25±0.58 | 20.68±0.43 | 21.96±1.36 | <0.001** |

** Significantly different from two groups at $p < 0.05$.

specialist [39]. They were then informed and familiarized with the FEOL and BHS protocols, the CMJ jump and the 30m sprint test. On each subsequent Friday from 15:00–17:00, participants returned and were instructed to perform standardized warm-up exercises, including joint mobility, dynamic stretching, mid-zone activation, and specific exercises (e.g., weighted half squat and sprint) before performing tests and training sessions [40]. All tests and training interventions were set up with an appropriate movement control mode. Technical feedback from the field was provided by experienced investigators. Tests and training sessions were repeated once the movements were performed incorrectly or not at maximum effort by the players.

## Measure

**Countermovement jump.**   Participants arrived at the lab and had a standardized warm-up followed by a 3-minute passive rest period. Subsequently, their CMJ performance was recorded twice (using participant rotation as the rest interval) and the better test was considered as the baseline for that training session. CMJ jump performance was measured again at 3, 6, 9, and 12 minutes after the participants performed one intensity of either FEOL or BHS protocol, respectively. Meanwhile, Participants used passive recovery between time points. Smart-Jump (Fusion Sport; Australia) was used to assess the CMJ performance of the players. CMJ jumps were performed using a uniform movement pattern: Maximal effort CMJs were performed using a self-selected depth and with hands-on-hips to prevent the influence of arm swing [41]. An investigator provided technical feedback to ensure that the jumps were performed correctly and encouraged all participants to make their best efforts in the test. The following variables were selected for data analysis: jump height (cm), jump peak power output (W), and jump impulse (Ns). Smart jump obtained the following formula by measuring the air time during the jump during the jump:

Jump height (cm) = (Flight time /1000)$^2$*g*100/8, where the unit of flight time is (ms), gravity (g) = 9.81 m/s.

Jump PPO(W) = 60.7*jump height+45.3*mass-2055, the PPO is an estimate and not a measurement.

**30m sprint test.**   Participants arrived at the laboratory and had a standardized warm-up followed by a 3-minute passive rest period. The performances of two rounds (with participant rotation as the rest interval) of 30m sprints of the participants were recorded and the better one was the baseline for the training session. Participants performed the FEOL or BHS protocol and 30m sprint performance was measured again at minutes 3, 6, 9, and 12, respectively. They were allowed to use a passive recovery mode between time points. The Brower timing system (TC-1H, USA) was placed at the starting line and the finish line of the 30 m sprint track, 1 m above the ground. Specific positions of the players were designated less than 0.5 m from the starting line with their feet apart in front of them (high starting position), arms at the side of the body, and hips and knees slightly flexed. On the GO signal, the participants were instructed to perform a sprint and the sprint time of 30 m (0.00s) was recorded.

## PAPE inducing action

**Flywheel eccentric overload (FEOL) exercise protocol.**   The FEOL protocol was accomplished by performing repetitive half squats using a flywheel ergometer (Exxentric AB, Sweden). The flywheel dynamometer includes 3 inertial moment intensities: low (inertia moment = 0. 015 kg·m$^2$), medium (inertia moment = 0.035 kg·m$^2$), and high (inertia moment 0.075 kg·m$^2$), and only one intensity is executed per training session, each intensity is performed for 3 sets × 6 reps with 2 min of passive recovery between each set [40]. Participants

started in a half squat position and were asked to execute the concentric phase as quickly as possible and to control the braking phase until the knees were flexed to approximately 90 degrees. An investigator provided technical feedback on each repetition. Participants received strong standardized encouragement to perform each repetition to the maximum extent possible [40].

**Loaded barbell half squats (BHS) protocol.** The BHS protocol was completed by performing weighted half squats using a barbell squat rack (YANBO J009, China). The BHS protocol consisted of 3 intensities: 40%, 60%, and 80% of 1RM. Only one intensity was performed in each training session, performing 5 sets × 3 reps with a 2-minute rest interval between sets for passive recovery [42]. Depending on the participant's height, an elastic band was attached to the barbell squat rack crossbar and used as a control brake for the eccentric phase of the squat until the knees were bent to approximately 90 degrees (hips touching the elastic band). Participants used a high bar position mode of half squat and were asked to descend slowly during the eccentric phase and finish as quickly as possible during the centripetal phase. Two investigators provided technical feedback for each repetition. Participants received strong standardized encouragement to perform each repetition to the maximum extent possible. All experimental measurements were conducted in the digitalized strength and conditioning laboratory of Guangdong Province, China.

## Statistical analysis

Statistical analysis was performed on SPSS software (v24.0, Chicago, USA). Data are presented as mean ± SD. Within-session reliability of test measures computed using a single measures two-way random intraclass correlation coefficient (ICC (2,1)) with an absolute agreement, inclusive of 95% confidence interval (relative reliability) [43] and the coefficient of variation (CV) (absolute reliability) [44]. ICC values less than 0.5 are indicative of poor reliability, values between 0.5 and 0.75 indicate moderate reliability, values between 0.75 and 0.9 indicate good reliability and values greater than 0.90 indicate excellent reliability. A CV ≤10% was set as the criterion to declare a variable as reliable. Analysis of variance (ANOVA) was performed for each variable in order to evaluate the significant modification of each score. Univariate analysis was performed using the groups (FEOL vs. BHS) as the between factor of the analysis and using the stage-I and stage-II as the within factor of the analysis. Multivariate analysis of variance for repeated measures (RM-MANOVA), was used to evaluate significant differences between the two groups (FEOL vs. BHS, named protocol), among different time points (0, 3, 6, 9 and 12 min, named Time), and in the interaction, Protocol × Time. RM-MANOVA was also used to evaluate significant differences among the different training intensity (low, medium, and high intensity, named Training intensity), and in the interaction, Time × Training intensity. One-way repeated-measure ANOVA was used to assess the effects of the two protocols (I- or II-BHS vs. I- or II-FEOL) on CMJ and sprint performance. If a meaningful F value was found, a Bonferroni post hoc test could be applied for correction. Time × protocol interaction and Time × training intensity interaction effects were analyzed using Pillai's trace multivariate criteria. Cohen's formula was used for evaluating the effect size (ES) of all dependent variables, including rest duration, exercise protocols, and intensity [45]. The influence degree was calculated and interpreted as follows: trivial < 0.200, 0.2 ≤ small < 0.6, 0.6 ≤ moderate < 1.20, 1.20 ≤ large < 2.0, very large ≥2.000 [40]. The significance was set at $p < 0.05$, and confidence interval was set at 95%.

## Results

### Test reliability

Within-session reliability data are presented in Table 2, implying that all data reported moderate to high reliability, with the exception of the I-BHS group (ICC = 0.45) and the I-FEOL

**Table 2. Mean test ± standard deviations (SD), test reliability.**

| Stage | Groups | Metrics | Intensities | M±SD | ICC (2,1) (95%CI) | CV (%,95%CI) |
|---|---|---|---|---|---|---|
| Stage 1 | I-BHS | CMJ | Low | 50.62±1.19 | 0.88 (0.47–0.99) | 2.35 (0.18–4.52) |
| Stage 1 | I-FEOL | CMJ | Low | 55.16±1.24 | 0.80 (0.44–0.98) | 2.24 (0.17–4.31) |
| Stage 1 | I-BHS | CMJ | Medium | 52.28±0.92 | 0.89 (0.63–0.99) | 1.76 (0.14–3.39) |
| Stage 1 | I-FEOL | CMJ | Medium | 55.45±1.89 | 0.74 (0.30–0.97) | 3.40 (0.27–6.54) |
| Stage 1 | I-BHS | CMJ | High | 52.45±0.39 | 0.82 (0.48–0.99) | 0.75 (0.06–1.44) |
| Stage 1 | I-FEOL | CMJ | High | 55.84±1.80 | 0.76 (0.33–0.98) | 3.22 (0.25–6.19) |
| Stage 1 | I-BHS | PPO | Low | 4569.90±67.38 | 0.96 (0.80–1.00) | 1.47 (0.12–2.83) |
| Stage 1 | I-FEOL | PPO | Low | 4317.15±75.07 | 0.76 (0.37–0.98) | 1.74 (0.14–3.34) |
| Stage 1 | I-BHS | PPO | Medium | 4669.71±62.24 | 0.96 (0.81–1.00) | 1.34 (0.10–2.58) |
| Stage 1 | pre- FEOL | PPO | Medium | 4311.66±95.30 | 0.68 (0.28–0.97) | 2.21 (0.17–4.25) |
| Stage 1 | I-BHS | PPO | High | 4684.84±23.80 | 0.95 (0.84–1.00) | 0.50 (0.04–0.98) |
| Stage 1 | I-FEOL | PPO | High | 4358.38±109.21 | 0.77 (0.34–0.98) | 2.50 (0.20–4.81) |
| Stage 1 | I-BHS | Ns | Low | 246.93±3.39 | 0.97 (0.83–1.00) | 1.37 (0.11–2.64) |
| Stage 1 | I-FEOL | Ns | Low | 219.42±2.48 | 0.81 (0.45–0.98) | 1.13 (0.09–2.17) |
| Stage 1 | I-BHS | Ns | Medium | 251.36±2.48 | 0.98 (0.89–1.00) | 1.00 (0.08–1.90) |
| Stage 1 | I-FEOL | Ns | Medium | 220.04±3.75 | 0.81 (0.40–0.98) | 1.70 (0.13–3.28) |
| Stage 1 | I-BHS | Ns | High | 251.96±0.95 | 0.98 (0.92–1.00) | 0.38 (0.03–0.73) |
| Stage 1 | I-FEOL | Ns | High | 221.42±4.31 | 0.76 (0.34–0.98) | 1.94 (0.15–3.74) |
| Stage 1 | I-BHS | Sprint | Low | 4.20±0.06 | 0.62 (0.20–0.96) | 0.15 (0.12–2.87) |
| Stage 1 | I-FEOL | Sprint | Low | 4.00±0.03 | 0.63 (0.22–0.96) | 0.69 (0.05–1.33) |
| Stage 1 | I-BHS | Sprint | Medium | 4.15±0.04 | 0.45 (0.07–0.93) | 1.07 (0.08–2.07) |
| Stage 1 | I-FEOL | Sprint | Medium | 4.00±0.04 | 0.37 (0.03–0.91) | 0.91 (0.07–1.75) |
| Stage 1 | I-BHS | Sprint | High | 4.09±0.03 | 0.67 (0.27–0.97) | 0.79 (0.06–1.51) |
| Stage 1 | I-FEOL | Sprint | High | 4.03±0.07 | 0.34 (0.02–0.90) | 1.65 (0.13–3.17) |
| Stage 2 | II-FEOL | CMJ | Low | 52.44±1.35 | 0.90 (0.62–0.99) | 2.58 (0.20–4.96) |
| Stage 2 | II-BHS | CMJ | Low | 54.29±0.95 | 0.54 (0.14–0.95) | 1.75 (0.14–3.36) |
| Stage 2 | II-FEOL | CMJ | Medium | 54.30±1.44 | 0.86 (0.46–0.99) | 2.64 (0.21–5.08) |
| Stage 2 | II-BHS | CMJ | Medium | 54.74±0.97 | 0.78 (0.37–0.98) | 1.77 (0.14–3.41) |
| Stage 2 | II-FEOL | CMJ | High | 55.04±1.31 | 0.79 (0.32–0.98) | 2.37 (0.19–4.56) |
| Stage 2 | II-BHS | CMJ | High | 55.00±0.57 | 0.87 (0.60–0.99) | 1.03 (0.08–1.99) |
| Stage 2 | II-FEOL | PPO | Low | 4683.90±82.19 | 0.96 (0.81–1.00) | 1.75 (0.14–3.37) |
| Stage 2 | II-BHS | PPO | Low | 4264.17±57.63 | 0.59 (0.18–0.96) | 1.35 (0.11–2.60) |
| Stage 2 | II-FEOL | PPO | Medium | 4796.98±87.18 | 0.95 (0.74–1.00) | 1.82 (0.14–3.49) |
| Stage 2 | II-BHS | PPO | Medium | 4350.42±47.76 | 0.53 (0.08–0.95) | 1.10 (0.09–2.11) |
| Stage 2 | II-FEOL | PPO | High | 4842.18±79.29 | 0.94 (0.69–1.00) | 1.64 (0.13–3.15) |
| Stage 2 | II-BHS | PPO | High | 4336.88±36.13 | 0.55 (0.09–0.95) | 0.83(0.07–1.60) |
| Stage 2 | II-FEOL | Ns | Low | 252.01±3.27 | 0.97 (0.88–1.00) | 1.30 (0.10–2.50) |
| Stage 2 | II-BHS | Ns | Low | 217.78±1.91 | 0.80 (0.45–0.98) | 0.88 (0.07–1.68) |
| Stage 2 | II-FEOL | Ns | Medium | 256.43±3.42 | 0.97 (0.84–1.00) | 1.33 (0.10–2.56) |
| Stage 2 | II-BHS | Ns | Medium | 222.81±4.26 | 0.53 (0.09–0.95) | 1.91 (0.15–3.68) |
| Stage 2 | II-FEOL | Ns | High | 258.14±3.08 | 0.97 (0.83–1.00) | 1.19 (0.09–2.29) |
| Stage 2 | II-BHS | Ns | High | 221.27±3.61 | 0.46 (0.03–0.94) | 1.63 (0.13–3.14) |
| Stage 2 | II-FEOL | Sprint | Low | 4.21±0.06 | 0.86 (0.54–0.99) | 1.53 (0.12–2.94) |
| Stage 2 | II-BHS | Sprint | Low | 4.01±0.03 | 0.77 (0.40–0.98) | 0.83 (0.07–1.60) |
| Stage 2 | II-FEOL | Sprint | Medium | 4.10±0.06 | 0.66 (0.23–0.97) | 1.54 (0.12–2.96) |
| Stage 2 | II-BHS | Sprint | Medium | 4.00±0.03 | 0.85 (0.51–0.99) | 0.79 (0.06–1.52) |

*(Continued)*

**Table 2.** (Continued)

| Stage | Groups | Metrics | Intensities | M±SD | ICC (2,1) (95%CI) | CV (%,95%CI) |
|-------|--------|---------|-------------|------|-------------------|--------------|
| Stage 2 | II-FEOL | Sprint | High | 4.07±0.05 | 0.58 (0.16–0.95) | 1.27 (0.10–2.45) |
| Stage 2 | II-BHS | Sprint | High | 3.96±0.05 | 0.01 (-0.07–0.55) | 1.27 (0.10–2.45) |

ICC values > 0.9 = excellent, 0.75–0.9 = good, 0.5–0.75 = moderate, and < 0.5 = poor, in accordance with Koo and Li. CV values were considered acceptable if < 10%.

group (ICC = 0.37) for stage-I of the medium-intensity post-training sprint test, the I-FEOL group (ICC = 0.34) for the high-intensity post-training sprint test, and the II-FEOL group (ICC = 0.01) for stage-II of the high-intensity post-training sprint test.

## Jump height

In stage-I, jump heights did not change significantly at low(FEOL = 0.015kg·m², BHS = 40% 1RM), medium(FEOL = 0.035kg·m², BHS = 60%1RM), and high(FEOL = 0.075kg·m², BHS = 80%1RM) training intensities, and the effect size of improvement for all dependent variables was below 0.60 (Fig 2A). Simultaneously, the jump height of the I-BHS and I-FEOL groups did no difference exists at each training intensity (Fig 2B).

Repeated measures ANOVA was performed on jump heights at each interval in the I-BHS group and the I-FEOL group. A 2-way ANOVA showed a significant main effect of rest intervals on jump height at low (FEOL = 0.015kg·m², BHS = 40%1RM), medium (FEOL = 0.035kg·m², BHS = 60%1RM), and high(FEOL = 0.075kg·m², BHS = 80%1RM) training intensities ($p < 0.05$). Mauchly's spherical hypothesis test showed that the interaction of the effect of Time × protocol on jump height had no statistical significance ($F_{low}$ = 1.917, $p = 0.190$, df = 4) at a low (FEOL = 0.015kg·m², BHS = 40%1RM) training intensity. However, it had a significant effect on jump height at moderate(FEOL = 0.035kg·m², BHS = 60%1RM) and high(FEOL = 0.075kg·m², BHS = 80%1RM) training intensities ($F_{medium}$ = 3.584, $p = 0.020$, df = 4; $F_{high}$ = 3.809, $p = 0.016$, df = 4; Fig 2A). In medium (FEOL = 0.035kg·m², BHS = 60%1RM) training intensity, the jump height at 3 (9.12 ± 15.84% increase, ES = 0.472), 6 (6.03 ± 12.97% increase, ES = 0.389), and 9 min (6.20 ± 10.07% increase, ES = 0.513) were increased in I-FEOL group compared with the I-BHS protocol. And in high (FEOL = 0.075kg·m2, BHS = 80%1RM) training intensity, the jump height at 3 (7.78 ± 9.90% increase, ES = 0.561), 6 (8.96 ± 12.15% increase, ES = 0.579), and 9 min (8.78 ± 11.23% increase, ES = 0.608) were enhanced in I-FEOL group compared with I-BHS group (Fig 2A, $F_{high}$ = 3.044, $p = 0.037$, df = 4). Furthermore, the interaction of Time × training barely affected jump height ($p > 0.05$, Fig 2B). Additionally, repeated-measures ANOVA was performed on the jump height at each rest interval at different intensities. None of the effects of the interaction of Time × training intensities on jump height had any statistical significance ($p > 0.05$, Fig 2B).

In stage-II, different intensities of BHS and FEOL training also did not affect the jump height (Fig 2A and 2B). Repeated measures ANOVA of jump height at each rest interval in the II-FEOL and II-BHS groups showed that the Time × protocol interaction did not affect jump height at low(FEOL = 0.015kg·m², BHS = 40%1RM) and medium(FEOL = 0.035kg·m², BHS = 60%1RM) training intensities ($F_{low}$ = 1.689, $p = 0.185$, df = 4; $F_{medium}$ = 0.923, $p = 0.580$, df = 4; Fig 2A). However, a significant interaction effect (Time × protocol) was observed at high training intensities ($F_{high}$ = 3.044, $p = 0.037$, df = 4; Fig 2A). Furthermore, the interaction of Time × training barely affected jump height ($p > 0.05$, Fig 2B).

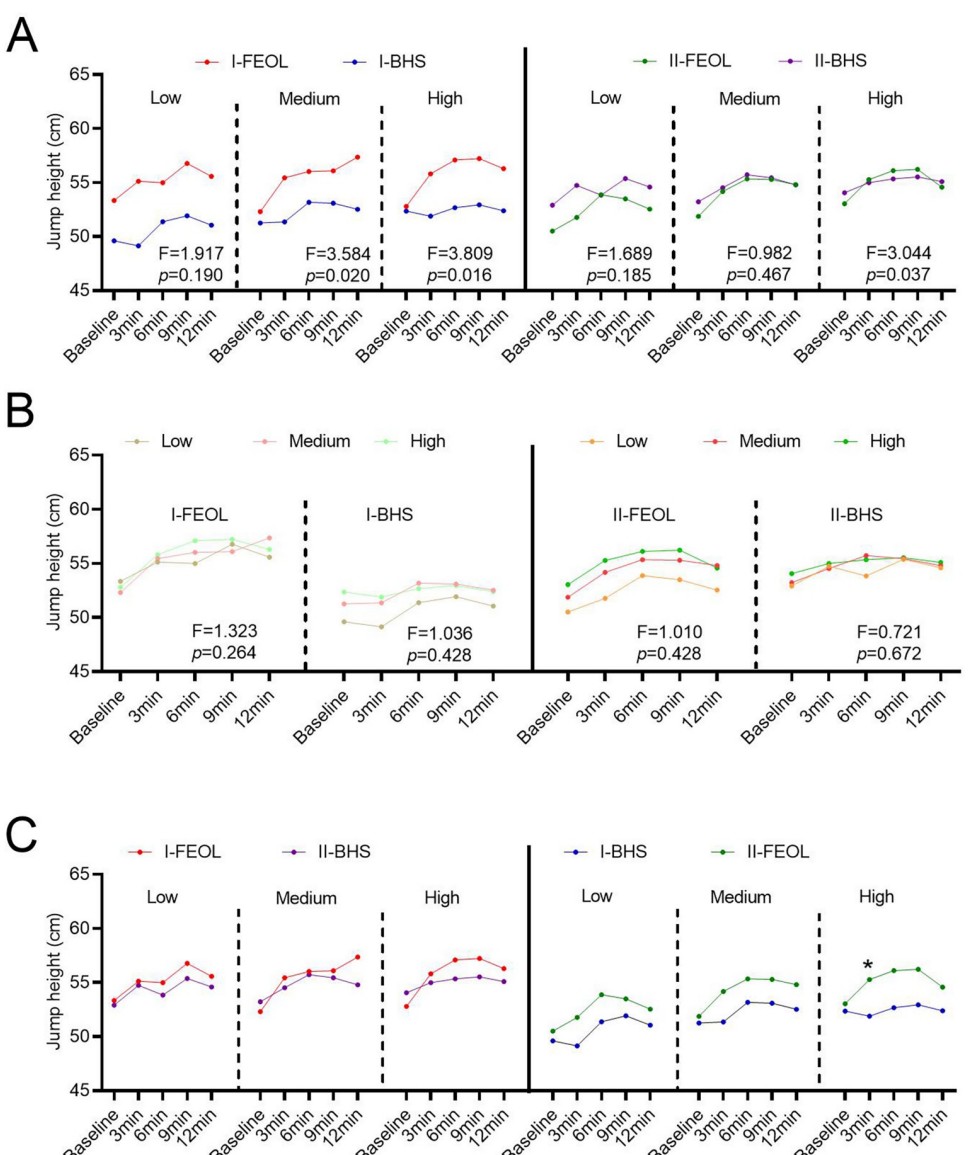

**Fig 2. Effects of training intensity, interval time, and training method on jump height in two stages.** (A) Time × protocol interaction effects on jump height over time in different training intensities. (B) Time × training intensity interaction effects on jump height over time under different training methods. (C) Jump height of the same population receiving different training protocols in two stages.

For participants who underwent FEOL training in stage-I, BHS training in stage-II did not affect the jump height among different intensities at 3 (ES = 0.084), 6 (ES = 0.243), 9 (ES = 0.202), and 12 (ES = 0.208) min (p > 0.05, Fig 2C). Similarly, for the participants who underwent BHS training in stage-I, FEOL training in stage-II did not affect the jump among different intensities at 3 (ES = 0.432), 6 (ES = 0.283), 9 (ES = 0.338), and 12 (ES = 0.314) min (p > 0.05, Fig 2C).

## Jump peak power

In stage-I, there was no significant difference after BHS and FEOL training at the three training intensities, and the effect size of improvement for all dependent variables ranged from 0.072–0.455 (Fig 3A). Additionally, there was no significant difference in PPO between I-BHS

and I-FEOL group at different measuring time points (Fig 3A). Furthermore, there was no significant difference in PPO between I-BHS and I-FEOL group at each training intensity (Fig 3A).

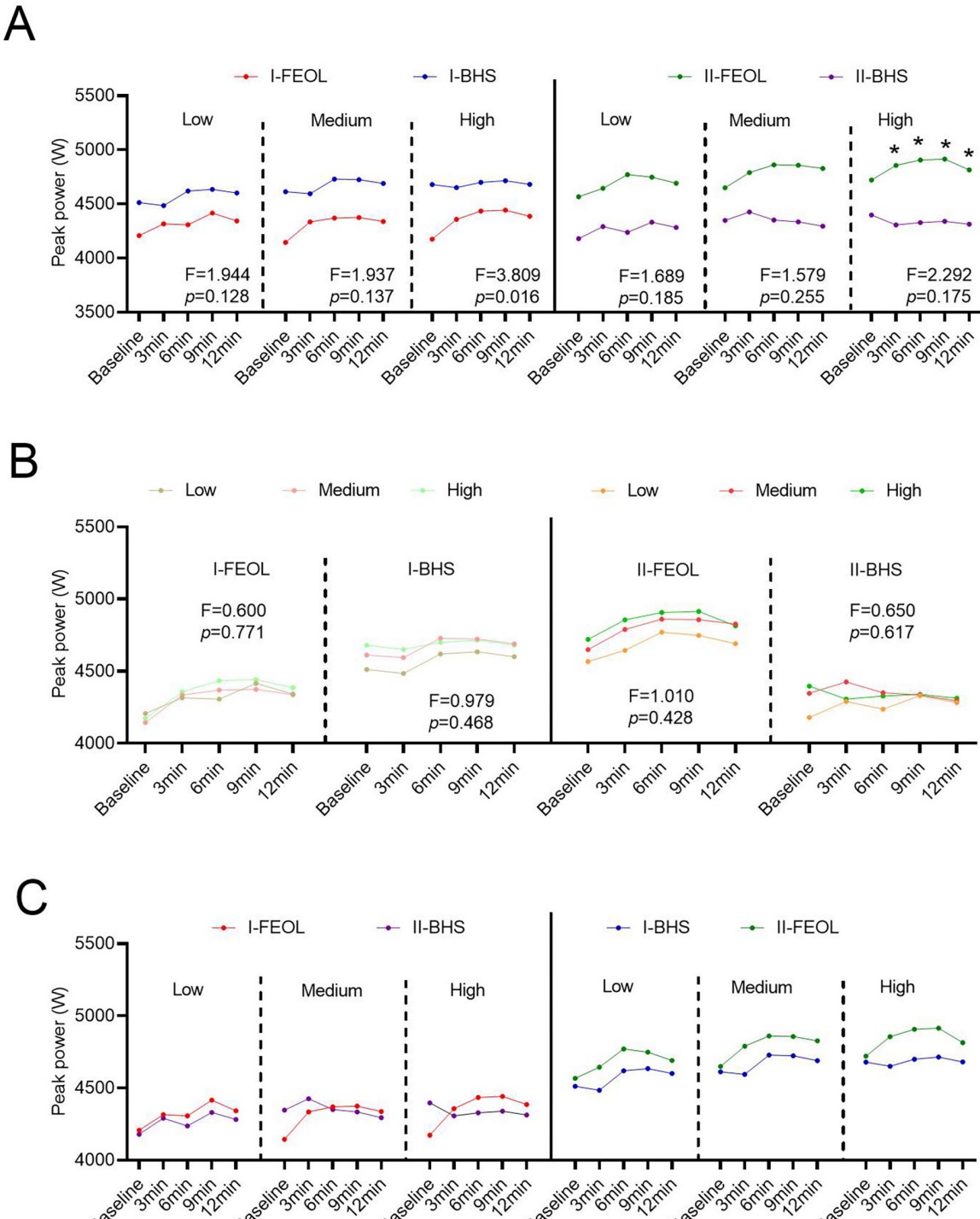

**Fig 3. Effects of training intensity, interval time, and training method on jump peak power in two stages.** (A) Time × protocol interaction effects on PPO over time in different training intensities. (B) Time × training intensity interaction effects on PPO over time with different training methods. (C) Comparison of PPOs of the same crowd under different training methods in the two stages.

The 2-way ANOVA showed a significant main effect of rest interval on PPO ($p < 0.05$). At high(FEOL = 0.075kg·m$^2$, BHS = 80%1RM) training intensity, the interaction of Time × protocol had a significant improving effect on PPO ($F_{high}$ = 3.809, p = 0.016, df = 4; Fig 3A). However, the interaction of Time × training at intensity also did not affect PPO ($p > 0.05$, Fig 3B).

In stage-II, different intensities of BHS and FEOL training did not affect PPO (Fig 3A and 3B). Furthermore, there was no significant change in PPO at the levels of Time × protocol interaction and Time × training intensity interaction ($p > 0.05$, Fig 3A and 3B).

For participants who underwent FEOL training in stage-I, BHS training in stage-II nearly did not affect PPO ($p > 0.05$, Fig 3C). Additionally, for the participants who underwent BHS training in stage-I, FEOL training in stage-II did not affect the PPO at each time point ($p > 0.05$, Fig 3C).

## Jump impulse

In stage-I, the jump impulse after BHS and FEOL training was almost unchanged compared with the baseline value (Fig 4A). However, the impulse in the I-BHS group was significantly higher than that in the I-FEOL group at 9 min, and 12 min at low (FEOL = 0.015kg·m$^2$, BHS = 40%1RM), medium (FEOL = 0.035kg·m$^2$, BHS = 60%1RM), and high (EFOL = 0.075kg·m$^2$, BHS = 80%1RM) intensities ($p < 0.05$; Fig 4A). At medium (FEOL = 0.035kg·m$^2$, BHS = 60%1RM) and high (FEOL = 0.075kg·m$^2$, BHS = 80%1RM) training intensities, the Time × protocol interaction affected the impulse (Fig 4A). Compared to the I-FEOL protocol, the impulse in I-BHS group was markedly higher at medium ($F_{medium}$ = 3.775, p = 0.016, df = 4) and high training intensities ($F_{high}$ = 3.042, p = 0.037, df = 4).

In stage-II, the impulse of the II-FEOL group was significantly higher than that of the II-BHS group in 6 min, 9 min, and 12 min at low (FEOL = 0.015kg·m$^2$, BHS = 40%1RM), medium (FEOL = 0.035kg·m$^2$, BHS = 60%1RM), and high (EOL = 0.075kg·m$^2$, BHS = 80%1RM) intensities ($p < 0.05$; Fig 4A). Neither the Time × protocol interaction term nor Time × training intensity showed an effect on impulse (Fig 4A and 4B).

For participants who performed FEOL training in stage-I, there was no significant difference in the impulse between both stages after receiving BHS training in stage-II ($p > 0.05$, Fig 4C). Similarly, for participants who performed BHS training in stage-I, there was no significant difference in the impulse between the two stages after receiving FEOL training in stage-II ($p > 0.05$, Fig 4C).

## Sprint speed

In stage-I, the sprint speed after BHS and FEOL training almost had no significant change compared to the baseline value ($p > 0.05$, Fig 4A). The sprint speed of the I-FEOL group was significantly lower than that of the I-BHS group at 9 min of low (3.81 ± 1.26% decrease, p = 0.009, ES = 0.843; Fig 5A) and medium (2.74 ± 1.64% decrease, p = 0.029, ES = 0.759; Fig 5A) intensity. The Time × protocol interaction and the Time × training intensity interaction barely affected the sprint speed ($p > 0.05$, Fig 5A and 5B).

In stage-II, there was no significant difference in sprint speed between the II-FEOL group and the II-BHS group at each rest interval at low (FEOL = 0.015kg·m$^2$, BHS = 40%1RM) and medium (FEOL = 0.035kg·m$^2$, BHS = 60%1RM) training intensities. However, at high (FEOL = 0.075kg·m$^2$, BHS = 80%1RM) training intensity, the sprint speed of the II-FEOL group was significantly higher than that of the II-BHS group at 6 min (2.81 ± 1.17% increase, p = 0.042, ES = 0.73), 9 min (2.55 ± 0.38% increase, p = 0.032, ES = 0.75), and 12 min (3.16 ± 1.29% increase, p = 0.019, ES = 0.791) (Fig 5A). Although there was no significant

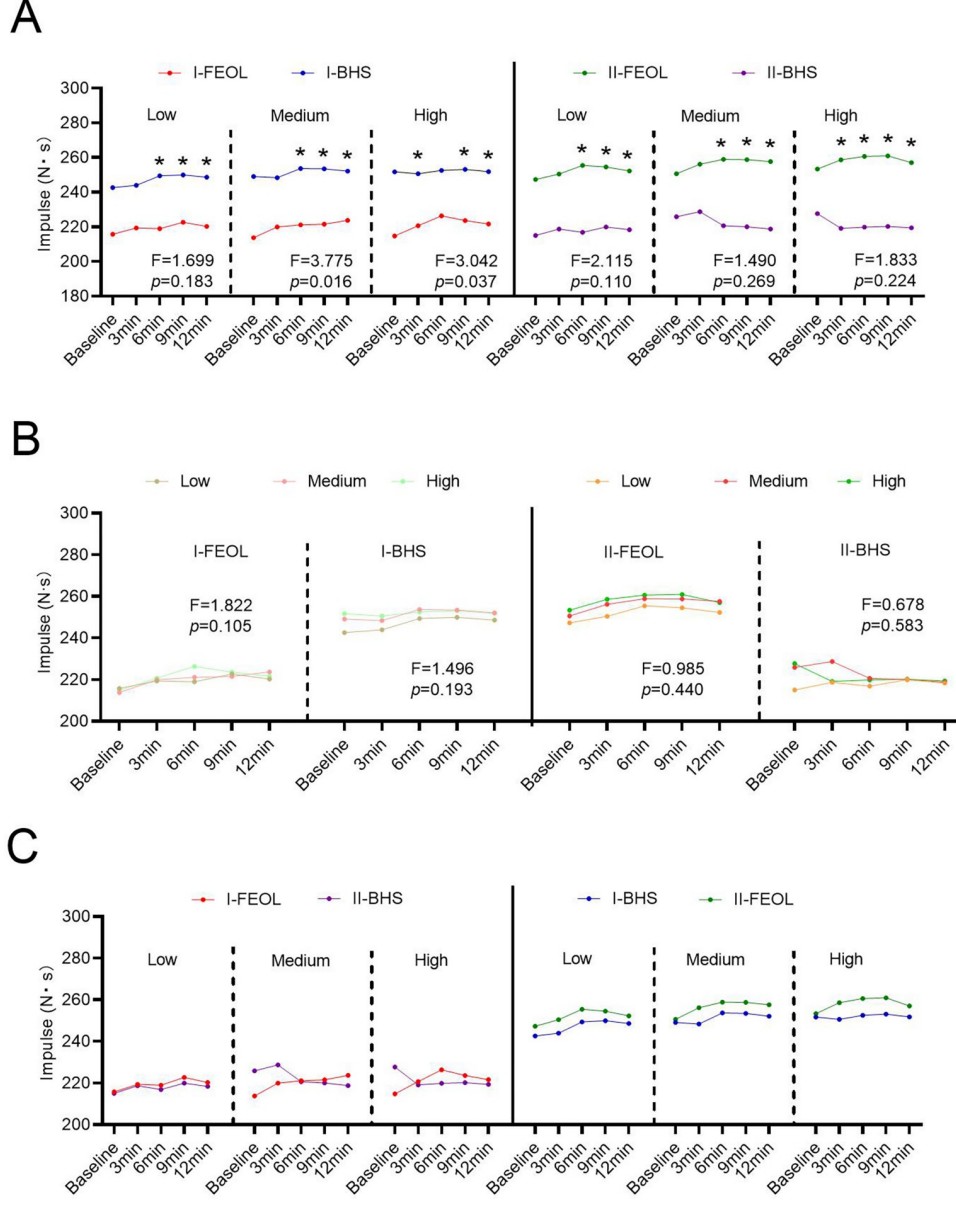

**Fig 4. Effects of training intensity, interval time, and training method on jump impulse in two stages.** (A) Time × protocol interaction effects on impulse over time in different training intensities. (B) Time × training intensity interaction effects on impulse over time in different training methods. (C) Comparison of impulses of the same crowd under different training methods in the two stages.

difference in sprint speed between the II-FEOL and II-BHS groups at the level of Time × protocol interaction, there was a significant difference in sprint speed between the II-BHS and II-FEOL groups at the level of Time × training intensity (F = 2.500, p = 0.029; Fig 5B).

For participants who underwent FEOL training in stage-I, BHS training in stage-II had no effect on sprint speed (p > 0.05, Fig 5C). Similarly, for participants who underwent BHS training in stage-I, FEOL training in stage-II did not affect sprint speed (p > 0.05, Fig 5C).

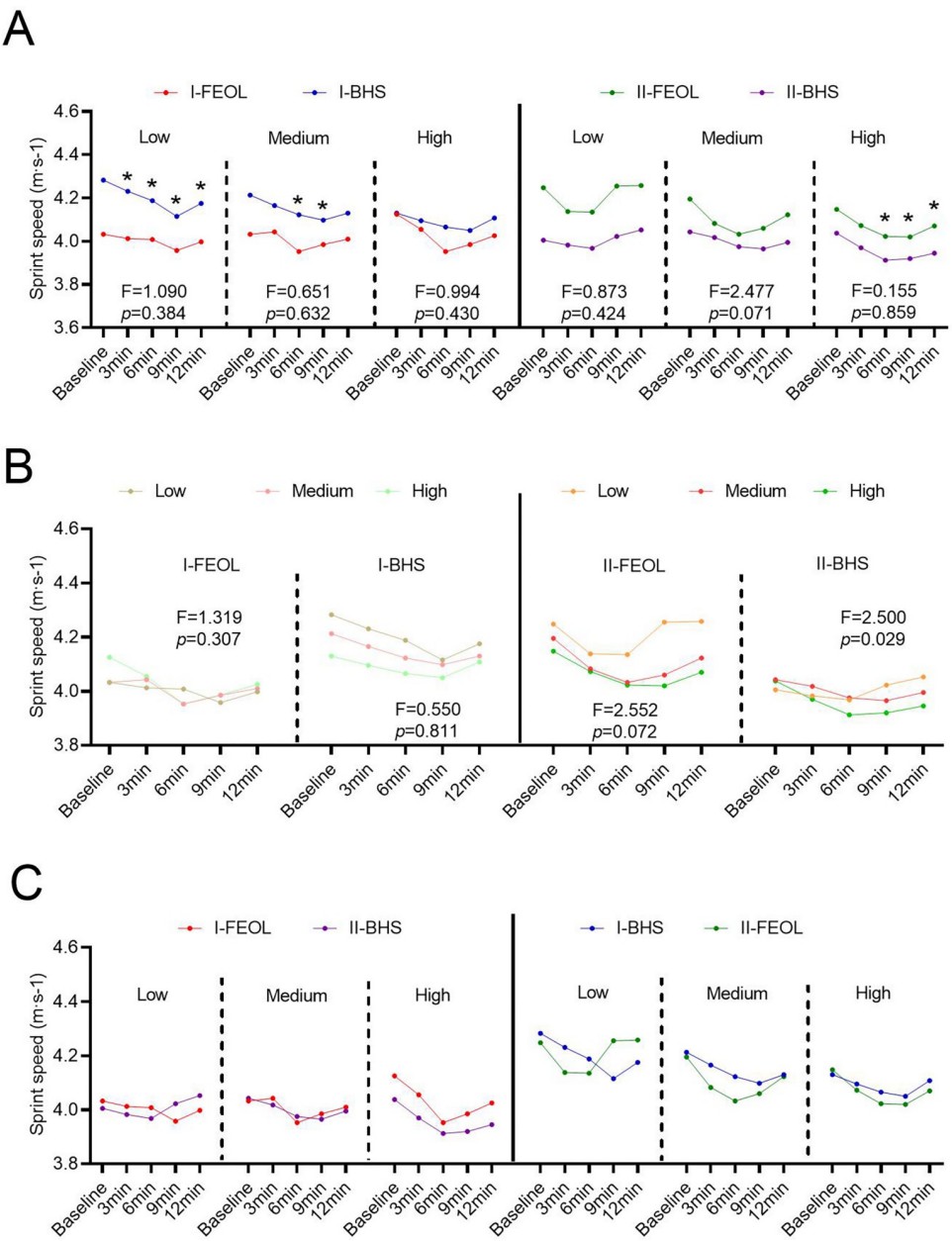

**Fig 5. Effects of training intensity, interval time, and training method on the 30-meter sprint in two stages.** (A) Time × protocol interaction effects on 30 m sprint speed over time in different training intensities. (B) Time × training intensity interaction effects on 30 m sprint speed over time with different training methods. (C) Comparison of 30 m sprint speeds of the same crowd under different training methods in the two stages.

## Discussion

The main finding of this study was that no difference exists on CMJ and 30m sprint PAPE were induced by BHS and FEOL training after a two-phase crossover trial. However, jump height at rest intervals of 3 (7.78 ± 9.90% increase, ES = 0.561), 6 (8.96 ± 12.15% increase, ES = 0.579), and 9 (8.78 ± 11.23% increase, ES = 0.608), were greater for FEOL than BHS protocol.

A 13-week half squat training program was conducted along well as the standardization of training frequency, time, sets, repetitions, and rest intervals, to ensure comparability of training conditions between the two groups in both phases. Under such a circumstance, it is necessary and challenging to control for these variables when comparing the results of different training programs [46]. Therefore, a randomized crossover design was adopted in our study. A crossover design has an advantage of requiring a small sample size and eliminating difference between individuals [47]. In our crossover design, athletes underwent a 2-week washout period between stage-I and stage-II, which was essential for adjusting the heterogeneity across the athletes enrolled in this study. Although they did not perform resistance training during this period, whether the effects of this specific training impact stage-II needs to be further explored.

Flywheel resistance training used in this study is indeed really different from the plyometric mode of muscle action and therefore relies not on the same mechanical (e.g. paralelle or serie elastic component) and neural elements (e.g. stretch reflex). The rotational inertia during flywheel exercise causes a greater eccentric overload than that produced by the traditional resistance exercise [22]. The barbell training under different intensities was constant resistance training. Flywheel, by contrast, provided unlimited resistance under every intensity and presented maximum or closely-maximum activation from the beginning due to its inertia force. The characteristics of flywheel inertia are that as the inertia increases, the velocity of peak concentric, peak eccentric, average concentric, and average eccentric tends to decrease, whereas the ratio of peak eccentric and peak concentric power increases [30]. However, we found that there was no difference exists in jump height between the BHS and FEOL groups at different intensities within each time interval. Such variations might be explained by the actual program used for PAPE-induction, as mentioned previously.

In addition, FEOL training presented to be jump height-increasable, compared with BHS. Comparing II-BHS with II-FEOL, FEOL was more effective in increasing jump height, especially at 3 min, 6 min, and 9 min after training. Likewise, comparing I-BHS with I-FEOL, the latter was more effective at 3 min, 6 min, 9 min, and 12 min after training [26]. Previous studies demonstrated that a longer recovery interval (4 to 8 min) resulted in a better PAPE effect, compared with a shorter recovery time (2 to 3 min), though individual variations were observed therein [48]. Additionally, it was reported that the afferent excitability caused by muscle contraction could last for 3 to 10 min after exercise [49], which was consistent with our findings.

Both BHS training and FEOL training could not improve the performance of 30-meter sprint possibly due to different moving patterns of half-squat (BHS and FEOL) and sprint. However, the effect of preload stimulation on sprint performance was ambiguous. Some studies reported improvements [18, 50–52], some showed no change [19, 35, 36], and several proposed different results depending on testing distance [16] and timing of measuring [50]. This study found that the BHS group had lower sprint speed under high-intensity (BHS = 80% 1RM) sessions, especially at intervals of 9 min and 12 min, compared with low (BHS = 40% 1RM) and moderate (BHS = 60%1RM) intensities ($F$ = 2.500, $p$ = 0.029; Fig 5B). This difference might be associated with the distance (30 compared to the previous 5 and 10 meters) and measuring time.

There are still some limitations in this study. We assumed that the low, medium and high resistance were equivalent in the FEOL and BHS protocols. The results are therefore inherently based on how well the intensities are matched. However, due to the difference in the working patterns of the barbell and flywheel, the resistance is actually different at equal intensities. The effect of the PAPE method and recovery time on the PAPE enhancement effect was analyzed using repeated measurement, but the influence of muscle strength on the effect of PAPE was

not investigated. Furthermore, in light of the parameters of this study, neither BHS nor FEOL could significantly increase the jump height of basketball players, whereas jumping ability is the basic athletic quality required by many events [37, 42]. Whether this is related to training volume, training intensity, muscle fiber type, or other factors remains to be investigated. Furthermore, we took into account the greater risk of 30-m maximal sprint-induced muscle damage. However, the repeated bout effect associated with eccentric strength training influenced the magnitude and etiology of the experimental training's residual effects. Therefore, it was conceivable that the PAPE effects observed for sprint performance after 3 days of training did not represent the effects that could be observed after the first occurrence of eccentric strength training. In addition, gender factor must be considered in the practical application of PAPE [53]. Rixon et al. [54] enrolled 30 participants to perform maximal isometric squat protocol (maximal voluntary contraction PAPE) and maximal dynamic squat (DS) protocol (DS-PAPE) to induce PAPE, and the results showed that male participants were significantly better than females in both jump height and power output. Tsolakis et al. [53] conducted a study on 23 participants and showed that after inducing PAPE by isometric (3 sets of 3 sec) or plyometric (3 sets of 5 repetitions) bench and leg press, male participants outperformed female participants in both CMJ and bench press throw. As this study only analyzed the sports performance of male basketball players, we still need to expand the sample size and take gender into account in future research.

In summary, although barbell (BHS) and flywheel (FEOL) half squat training did not provide a significant PAPE effect on explosive power (CMJ and sprint) in male basketball players, FEOL training showed a better potential effect on enhanced CMJ jump performance following high-intensity (EOL = 0.075kg·m$^2$) training.

## Practical applications

Jumping performance is important for basketball players. However, as mentioned previously, the effect of PAPE is only transitory and could not be effective during an entire match. Furthermore, players run from the onset of the match which can therefore offset the effectiveness of a conditioning protocol on jumping performance. Therefore, the limit related to the activity would instead stand upon the practicability of this protocol. PAPE protocols induced by half squats (barbell or flywheel) should be implemented in training to improve jumping performance during exercise and thus improve training effectiveness.

## Supporting information

**S1 Data.**
(XLSX)

## Acknowledgments

The authors thank the participants for their participation in this study.

## Author Contributions

**Conceptualization:** Hezhi Xie, Wenfeng Zhang, Xing Chen, Jian Sun.

**Data curation:** Hezhi Xie, Wenfeng Zhang, Xing Chen, Jiaxin He, Junbing Lu, Yuhua Gao, Duanying Li, Guoxing Li, Hongshen Ji, Jian Sun.

**Formal analysis:** Junbing Lu, Yuhua Gao, Duanying Li, Guoxing Li, Hongshen Ji, Jian Sun.

**Funding acquisition:** Duanying Li.

**Investigation:** Hezhi Xie, Wenfeng Zhang, Xing Chen, Jiaxin He, Junbing Lu, Yuhua Gao.

**Methodology:** Hezhi Xie, Wenfeng Zhang, Xing Chen, Guoxing Li, Hongshen Ji.

**Project administration:** Hezhi Xie, Jian Sun.

**Resources:** Hezhi Xie, Xing Chen, Duanying Li, Guoxing Li.

**Software:** Wenfeng Zhang, Xing Chen, Junbing Lu, Yuhua Gao, Duanying Li, Guoxing Li, Hongshen Ji.

**Supervision:** Xing Chen, Hongshen Ji, Jian Sun.

**Validation:** Jiaxin He, Jian Sun.

**Visualization:** Jiaxin He, Yuhua Gao, Duanying Li, Guoxing Li, Hongshen Ji, Jian Sun.

**Writing – original draft:** Hezhi Xie, Wenfeng Zhang, Xing Chen, Jiaxin He, Junbing Lu, Yuhua Gao, Duanying Li, Guoxing Li, Hongshen Ji.

**Writing – review & editing:** Hezhi Xie, Wenfeng Zhang, Xing Chen, Jian Sun.

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
