## [Decision Letter · Decision Letter 0]

25 Mar 2022

PONE-D-22-02783Eccentric overload exercises versus loaded half squats for basketball players: which is better for induction of postactivation potentiation?PLOS ONE

Dear Dr. Sun,

Thank you for submitting your manuscript to PLOS ONE. After careful consideration, we feel that it has merit but does not fully meet PLOS ONE’s publication criteria as it currently stands. Therefore, we invite you to submit a revised version of the manuscript that addresses the points raised during the review process, and especially the important issues regarding statistical analyses.

We look forward to receiving your revised manuscript.

Kind regards,

Laurent Mourot

Academic Editor

PLOS ONE

Journal Requirements:

Reviewers' comments:

Reviewer's Responses to Questions

**Comments to the Author**

1. Is the manuscript technically sound, and do the data support the conclusions?

Reviewer #1: Partly

Reviewer #2: Partly

Reviewer #3: Yes

2. Has the statistical analysis been performed appropriately and rigorously? 

Reviewer #1: No

Reviewer #2: No

Reviewer #3: Yes

3. Have the authors made all data underlying the findings in their manuscript fully available?

Reviewer #1: Yes

Reviewer #2: Yes

Reviewer #3: No

4. Is the manuscript presented in an intelligible fashion and written in standard English?

Reviewer #1: No

Reviewer #2: Yes

Reviewer #3: Yes

5. Review Comments to the Author

Reviewer #1: General comment

The study aimed to test the influence of exercise modality (eccentric overloading or squatting) and intensity (light, moderate and heavy) on post-activation performance enhancement (and not PAP) in CMJ and sprint. The experimental protocol contain several bias that should be taken into account to appropriately discuss the findings. First, the authors implemented a cross-over designed for the conditioning exercise, but not for the load, which lead therefore to a progressive increase in load for all subjects. It would have therefore been more suitable to randomize the allocation order for load in order to avoid any adaptive effects from the load on the PAPE response. Then, the intensity and volume of the two conditioning protocols differed which could results into different workload between the two protocol. In absence of quantification of the total workload during exercise, it is difficult to account for this effect on the PAPE effect. Finally, this study thought to improve recommendation about conditioning protocol to improve jump performance in basketball players. However, the context of these recommendations should be precise in the context of this activity; that is, whether such protocol should be implemented at the onset of a match (but with likely confounding effects from sprint actions performed), or during training to optimize training effectiveness.

Furthermore, due to the multiple conditions (allocation group, exercise and intensity), the protocol and the results section are difficult to understand, and needs to be thoroughly amend or rephrase to improve this particularly limiting point. Many sentences are therefore confusing and scramble understanding of the results.

Finally, some elements are lacking to support statement provided particularly in the discussion section. Authors should therefore strengthen their rational in order to avoid sections that only describe findings of the present or previous studies, but provide more elements to understand the present findings and conclude on recommendations.

Please find below specific comments :

1. L 52: Undefined previously, please define

2. L 54-55: "Could increase PPO to a certain extent" is an inapropriate statement to present results. Please rephrase or present the magnitude of this increase.

3. L 58: performances were only assessed at different precise timing after exercise, and not during different period after exercise termination. Therefore, I suggest using time-points rather than time windows.

4. L 67-70: Please conform to the findings of the cited reference (and others e.g.; Blazevich and Babault, 2019; DOI: 10.3389/FPHYS.2019.01359), by mentioning the difference between electrically-evoked and voluntary improvements in performance. With reference to the area of interest of the present study, authors monitored PAPE and not PAP

5. L 72-74: Only referring to phosphorylation of MHC or change in fascicle pennation angle is really scarce in light of others potentials factors driving improvement in voluntary performance (e.g. muscle temperature, muscle blood flow), and particularly when focusing on eccentric mode of muscle action (e.g. neural factors). Please amend to provide a reliable overview of these mechanisms.

6. L 78: typing error : provided

7. L 80-81: Because squat exercise modality can induce a PAPE only in the lower limbs (in comparison to the upper limbs), I suggest the authors to precise this element in order to avoid misunderstanding by novice readers.

8. L 82-83: Please precise which parameters were improved in these studies (e.g. jump height, rate of force development, reaction time, maximal power, maximal speed, ...). Indeed, the conflicting findings about the factors improved by PAPE which depend upon the conditioning sequence should be avoid here.

9. L 85: "Has been pro" should be rephrase for a better understanding

10. L 96-100: Please also report the above-mentioned study (i.e. Norrbrand 2010) that reported similar improvement in MVC than the concentric training. A more reliable overview of the litterature, and the conflicting conclusions, should be drawn here.

11. L 101-103: Although scarce, please cite some references focusing on jumping performance (e.g. French et al., 2003, DOI: 10.1519/ 1533-4287(2003)017<0678:CIDEPF>2.0.CO;2)

12. L 116: It is unclear what are the short-term effects mentioned here. The authors stated a two-stages randomized cross-over study, lasting for 13 weeks. Therefore, are the short-term effects the acute effects following bouts of exercise, or the immediate effects of the training protocol tested at the end of the last bout? Should be precise.

13. L 116: The greater sprint performance could only be speculated here, and could therefore be mentioned as an expected finding at the end of the introduction section. I suggest the authors to rephrase this sentence in order to differentiate between the expected, and the actual findings of the protocol.

14. L 129: Rephrase as : and they were then informed about the EOL protocol.

15. L 130-131: Please provide additional explanations about the expected effects of the familiarization procedure for experienced subjects.

16. L 133-139: The procedure implemented to assess CMJ and sprint performance requires further supporting elements. Could the authors indicate for instance why they did not alternate the testing between CMJ and sprint? It could be suppose that the first 3 bouts of exercse (and particularly during the EOL protocol) induced some muscular impairments leading to bias the maximal performance during CMJ or sprint. Could the authors provide reliability measurements between sessions to account for this possible confounding factor?

17. L 139: Please add the time-delay allowed between the two sequences of this protocol.

18. Also, justify that the 1-week washout period is sufficient to alleviate the effects of the first protocol on the testing and effectiveness of the second sequence. (which seems unlikely).

19. L 149: I recommend the authors to avoid the excessive use of decimals and report the appropriate precision provided by the measurements they performed. For instance, rouding age to the nearest year as they ask to the participants, or report body-mass to the precision allowed by the body scale.

20. L 161: Should be : "subjects were asked to"

21. L 170: Please detail what are those "basic information"

22. L 178-180: Do the authors control for the reproducibility between the two trials to ensure the validity of the calculated mean? If so, please provide the accepted range between trials to ensure two maximal performance.

23. L 181: What was the recovery modality allowed to the subjects between the different time points?

24. L 191-193: Please provide the height of the photoelectric cells.

25. L 205: In accordance with one of my previous comment regarding the lack of details about the "basic information" recorded by means of body scale, the authors should provide additional information regarding the method used to calculate body surface area.

26. L 207-209: Please precise whether the pace was controlled during the downward phase of the half squat. If so, this information should be added to provide an estimate of the braking resistance applied by the subjects

27. L 214-215: It seems that the amount of repetitions per set is lacking here. Please amend.

28. L 225-227: Provide further explanations about the meaning one his one-way ANOVA. Indeed, your protocol implemented wo exercises and three modality. How could the authors therefore performed only a one-way ANOVA?

29. L 233-234: Statistical information are lacking in figures (significant different and not only ANOVA F or p values) forcing the readers to back-and-forth between figures and text. It could be therefore suitable to add the calculated 95%CI on the graphs and symbols to indicate statistical differences.

30. L 240-243: The comparisons between time points and intensity for pre-HS and pre-EOL is unclear here. It is state in the protocol that each session differed from the other by the testing protocol (HS or EOL) or the intensity (low, med and high). Please amend therefore to understand which variables were compared here.

31. L 249: Could you please indicate the degree of freedom for the F value of the ANOVA. In addition, I suggest to avoid the appelation of week1, 2 or 3, should rather precise the intensity that is one of the main independent variable.

32. L 252-253: This sentence requires rephrasing to avoid misinterpretation. Under the current version, this sentence would means that the performance at PRE is significantly related to measurements made at POST. However, it seems that this analysis was not performed, and is not of primary importance for this study.

33. L 263-265: It could be suitable to report the magnitude of the significant changes in the text to allow the reader a better adoption of the findings.

34. L 277-278: This third comparison lacks of support here, and should be presented in the statistics section.

35. L 282-283: This sentence is misconducting here since the mention of rest interval suggest that post-measurements would be presented, while it seems that comparison is performed on PRE measurements. Please amend and/ or correct.

36. L 284-285: Please state clearly this difference in the text

37. L 288-290: Again this sentence is misconducting, since is could be understand that the PRE-HS value is significantly correlated to PPO. Please rephrase.

38. L 333-335: How could the authors state that the sprint speed of PRE-EOL could be lower than PRE-HS at 9-min rest interval? Once again this sentence is conflicting here and needs to be rephrase.

39. L 342-344: Interactions rather revealed significant differences than influenced results. Please rephrase for a better understanding.

40. L 361-362: It would be suitable to differentiate between stretch shortening cycle occurring during running or CMJ for instance and the succession of eccentric and concentric phase with no (or only a few) storage of passive elastic energy (e.g. squatting). Flywheel resistance training used in the present study is indeed really different from plyometric mode of muscle action, and relies therefore not on the same mechanical (e.g. paralelle or serie elastic component) and neural elements (e.g. stretch reflex). Please amend and rephrase.

41. L 369-370: This aim differ from the scientific question addressed in the introduction section, and is therefore not supported by the rational. Please ensure consistency about the objectives of this study throughout the manuscript.

42. L 398-404: There are contradictory findings mentioned here between improvement in CMJ height and PPO due to EOL training, while non-statistical difference is noted. Please based your statement on the statistical findings and not graphical reading.

43. L 428-430: This section is quite descriptive and lacks of elements to explain the absence of significant improvement in this study. The authors should strenghten their discussion and provide scientific elements explaining their findings. Such approach would benefits to coaches in order to better understand how they could improve their intervention.

44. L438-440: I acknowledge that jumping performance is important for basketball players. However, as it is mentioned previously, the effect of PAPE is only transitory and would not be effective during an entire match. Furthermore, players runs from the onset of the match that can therefore offset the effectiveness of a conditioning protocol on jumping performance. Therefore the limit related to the activity would rather stands upon the practicability of this protocol. Authors should therefore provide recommendations or perspectives about how to applied such protocol for performance improvement. That is, should a PAPE protocol should be implement before a match (with a relatively low efficiency given the other actions, see above), or to used during training session in order to improve performance during exercise in order to improve training effectiveness?

45. L 447-449: This sentence contradicts the first one of the conclusion section. Please ensure coherence about the presence or absence of PAPE phenomenom in the present study. This question remains unclear while reading this section.

46. L 449-452: Both squatting and flywheel are based on eccentric-concentric contractions here. It appear therefore obvious to precise more specifically the different exercise modalities (e.g.; phase duratin, intensity, ...).

Reviewer #2: Abstract

• Line 46-47: not clear, rephrase and explain clearly the design.

• Lines 49-51: not useful in an abstract.

• Lines 52-58: there is no consistency between the methods and the results. The dependent parameters were not listed, nor the extent of the changes was reported. Please rewrite this section in accordance with what reported in the methods

• The last sentence makes no sense since every movement is a combination of concentric/eccentric actions.

Introduction

• Lines 103-106: It is not clear what “long season” means here. Please explain.

• Overall, why half-squat and not other squat variations? Please elaborate.

Methods

• Line 195: PAP or PAPE?

• The statistical analysis should be rerun: it is not clear to me why two different two-way analysis were performed instead of a three-way.

This is a major point that should be addressed. Therefore, I have stopped my review here, since the results should be rewritten accordingly.

Reviewer #3: General comments:

This study compared the post-activation performance enhancement induced by two forms of resistance exercise (flywheel and barbell half squats) and the effects of such training protocols over a period of weeks. The authors should be congratulated for performing a cross-over multiple week investigation. The study, and particularly the longitudinal nature, are a useful addition to the literature. However, there are a number of issues that I wish to raise. The most significant of these is the fact that it is not clear exactly what statistical analyses have been performed due to some confusion in the wording. Many assumptions or justifications inherent in the analyses are also missing. These factors make it difficult to follow and assess aspects of the Results and Discussion. I have mentioned some other specific comments, too.

Comment 1: The manuscript refers to post-activation potentiation (PAP) throughout – attributed to phosphorylation of myosin regulatory light chains. However, this ignores the recent body of literature questioning these mechanisms and time-courses and suggesting that post-activation performance enhancement would be a more appropriate name. Please see https://doi.org/10.3389/fphys.2019.01359 and related literature for example. The Introduction as a whole is quite short and missing some key literature.

Comment 2: The terms used to describe the exercises are quite vague. For example, an eccentric overload could be achieved in many different ways and a ‘loaded half squat’ could also be achieved in different ways. In fact, it could be argued that both exercises are ‘loaded half squats’.

Comment 3: The loads used for the barbell squat are individual-specific (a percentage of 1RM) but the flywheel inertias are not. This warrants further consideration or discussion.

Comment 4: In the statistical analysis section, it is stated that one-way repeated measures ANOVAs were performed. However, a few lines later on talk about interaction effects, which suggest that at least a two-way ANOVA was performed. It is therefore unclear what statistical analyses were performed – how many ANOVAs, what factors, and what conditions or intensities are in each. It is also mostly unclear whether the two phases are being tested separately or with all data combined (and why). The inherent assumptions in your statistical model also need to be considered and discussed. For example, are you assuming that the two lowest intensities (flywheel and barbell) are equivalent and the two highest are equivalent, etc.? Should you control for baseline values and/or session number or any other variables? You should also report confidence intervals (e.g. https://doi.org/10.1080/14763141.2020.1782555 ). Other recommendations in that editorial such as reporting exact p-values should also be followed. These factors make it difficult to follow and assess aspects of the Results and Discussion. Perhaps the Discussion would be easier to follow if it followed the narrative of the tests – i.e. one ANOVA at a time and discussing the main and interaction effects / post-hocs clearly.

Specific comments:

Authors: It is not clear how there can be three co-first authors.

Short title: Compared to what?

Data availability statement: I suggest that you upload the data alongside the manuscript for readers and reviewers in line with journal policy.

Abstract:

Line 52: PPO should be defined. Peak power output? It would also be useful to include some statistical results within the abstract.

Introduction:

Lines 74-75: This statement is too dramatic/ambitious and should be reduced in everity.

Line 95: Other studies have investigated the relationship between flywheel moment of inertia and velocity or power (concentric and eccentric) during flywheel half squats – see https://doi.org/10.1080/02640414.2020.1860472 This is also important on line 217 where the statement about ‘different components of muscle power’ is quite vague and could be supported more.

Lines 101-106: Although not on jumping sports, quite a few PAPE studies have focused on vertical or horizontal jumps. These have investigated the effect of factors such as the flywheel inertia, number of sets, etc. on flywheel PAPE. For some examples:

https://doi.org/10.1123/ijspp.2019-0476 (review paper)

https://doi.org/10.1519/JSC.0000000000003214 (effect of inertia)

https://doi.org/10.1123/ijspp.2019-0411 (effect of volume)

https://doi.org/10.3390/sports9010005 (effect of inertia and jump direction on ground reaction force parameters)

Methods:

Lines 118-120: It’s unclear how many sessions this is in total. How many familiarisation sessions were performed if any? See https://doi.org/10.1123/ijspp.2017-0282 but be aware of the use of magnitude based inference (e.g. https://doi.org/10.1080/14763141.2020.1782555 )

Line 148: Why 12 participants? This number should be justified. See https://doi.org/10.1525/collabra.33267

Line 161 and elsewhere: I suggest being consistent with ‘participants’ or ‘subjects’ – the section is called ‘Participants’ so it would be good to stick with this.

Lines 178 and 188: Why were two trials averaged? Why two and why an average?

Lines 185-186: How were these parameters calculated? There are multiple possible methods for some of them.

Line 192: What was the starting position for the sprint? In line with the timing gate or slightly behind?

Line 201: Why 3 sets x 6 reps? The volume PAPE paper mentioned earlier could be used here, but it is important to justify these choices.

Lines 208-209: Were participants instructed to resist throughout the eccentric phase or only in a certain portion of it?

Lines 232-233: These should be reworded with signs such as ‘greater than or equal to’ to ensure that there are no gaps. For example, 1.195 and 1.995 currently have no category.

Results:

I suggest adding tables to make the results clearer.

Discussion:

Line 368: ‘shown’ may be better than ‘proved’ – to show more uncertainty.

It would be good to re-summarise the main overall results early in the Discussion.

Lines 369-370: More justification is needed for why basketballers might be different to other populations and why results might not continue over a season.

Line 382: What is meant by ‘ground-lifting’?

Lines 384-387: Some of the studies I mentioned above (e.g. the effects of inertia on PAPE) may be useful here. Likewise, for lines 390-391 where the differential effects of peak power / velocity / force are discussed.

Line 397: Could this be controlled for within the statistical analysis?

Line 411: Was this ‘tendency’ significant? If not then it should not be discussed as an effect. The same applies in line 448 (if not then this should not be part of the conclusion).

Lines 418/420/421 – consistency needed around minutes / -minutes / min

6. PLOS authors have the option to publish the peer review history of their article (what does this mean?). If published, this will include your full peer review and any attached files.

Reviewer #1: No

Reviewer #2: No

Reviewer #3: No

---

## [Author Response · Author response to Decision Letter 0]

25 May 2022

Dear editor and reviewers

Regarding the manuscript ID (NO.PONE-D-22-02783) entitled " Eccentric overload exercises versus loaded half squats for basketball players: which is better for induction of postactivation potentiation?". Thank you for your letter and the reviewers' comments on our manuscript entitled " Eccentric overload exercises versus loaded half squats for basketball players: which is better for induction of postactivation potentiation?" (NO.PONE-D-22-02783) were evaluated. These comments were valuable and helpful to us in revising and improving the paper, as well as important guidance for our research. We have carefully studied these comments and made revisions, which we hope will be approved by all of you. The revised parts are marked in red in the thesis.and The main corrections in the paper and the responses to the reviewers are listed below.

Responds to Reviewer #1:

1. Response to comment: L 52:Undefined previously, please define.

Responds: We greatly appreciate the reviewers' comments. We have changed the definition of PPO to Peak power output (PPO). In L52

2. Response to comment: L 54-55:"Could increase PPO to a certain extent" is an inapropriate statement to present results. Please rephrase or present the magnitude of this increase. 

Responds: Thank you for your valuable comments. For the reviewers' valuable comments, we have revised the content to EOL training in the second stage increased PPO to a certain extent, especially at 3 min (p=0.046, ES=0.716). In L56-58

3. Response to comment: L 58:Performances were only assessed at different precise timing after exercise, and not during different period after exercise termination. Therefore, I suggest using time-points rather than time windows.

Responds: Thank you for your valuable comments. For the valuable comments of the reviewers. We have changed the time windows to time-points. In L63-64

4. Response to comment: L 67-70:Please conform to the findings of the cited reference (and others e.g.; Blazevich and Babault, 2019; DOI: 10.3389/FPHYS.2019.01359), by mentioning the difference between electrically-evoked and voluntary improvements in performance. 

Responds: We are very grateful for the reviewers' comments. For the reviewers' valuable comments, we add the study by Blazevich and Babault. In this study, the term PAPE was used in relation to the enhancement of muscle contraction after conditioning activity. In L80-81

5. Response to comment: L 72-74:Only referring to phosphorylation of MHC or change in fascicle pennation angle is really scarce in light of others potentials factors driving improvement in voluntary performance (e.g. muscle temperature, muscle blood flow), and particularly when focusing on eccentric mode of muscle action (e.g. neural factors). Please amend to provide a reliable overview of these mechanisms.

Responds: We are very grateful for the reviewers' comments. After carefully reading the reviewers' comments, we have added the concepts and mechanisms related to PAP and PAPE. In L83-90.

6. Response to comment: L 78:Typing error : provided 

Responds: Thank you for your valuable comments. After carefully reading the reviewers' comments, we have changed the typing to provided. 

7. Response to comment: L 80-81:Because squat exercise modality can induce a PAPE only in the lower limbs (in comparison to the upper limbs), I suggest the authors to precise this element in order to avoid misunderstanding by novice readers.

Responds: We greatly appreciate the reviewers' comments. For the reviewer's valuable comment, we have changed it to Squat exercise can induce a PAPE in the lower limbs. In L94.

8. Response to comment: L 82-83:Please precise which parameters were improved in these studies (e.g. jump height, rate of force development, reaction time, maximal power, maximal speed, ...). Indeed, the conflicting findings about the factors improved by PAPE which depend upon the conditioning sequence should be avoid here.

Responds: Thank you for your valuable comments. After carefully reading the reviewers' comments, we have listed the relevant metrics: strength, power, sprint, jumps and maximal voluntary contraction (MVC). In L100.

9. Response to comment: L 85:"Has been pro" should be rephrase for a better understanding

Responds: Thank you for your valuable input. After carefully reading the article, we have changed to Heavy and moderate weighted squats (90% and 60% of 1RM, respectively) have been reported to improve sprint and CMJ performance in male subjects. In L97

10. Response to comment:. L 96-100:Please also report the above-mentioned study (i.e. Norrbrand 2010) that reported similar improvement in MVC than the concentric training. 

Responds: Thank you for your valuable input. After carefully reading the article, we have made revision and addition in the corresponding places.

11. Response to comment: L 101-103:Although scarce, please cite some references focusing on jumping performance (e.g. French et al., 2003, DOI: 10.1519/ 1533-4287(2003)017<0678:CIDEPF>2.0.CO;2)

Responds: Thank you for your valuable comments. After carefully reading the reviewers' comments, we have described their impact on jump performance and added relevant references. In L114-116.

12.Response to comment: L 116:It is unclear what are the short-term effects mentioned here. The authors stated a two-stages randomized cross-over study, lasting for 13 weeks.

Responds: We are very grateful for the reviewers' comments. We have rewritten the sentences to compare the acute effects of the EOL protocol (three intensities) and the HS protocol (three intensities) on CMJ jump height or 30 m sprint performance after each training session(each intensity). In L129-131

13.Response to comment: L 116:The greater sprint performance could only be speculated here, and could therefore be mentioned as an expected finding at the end of the introduction section.

Responds: We are very grateful to the reviewers for giving such a high rating to our article. We have rewritten the Experimental approach to the problem in order to be more clear about the design of the experiment and in response to the question you raised. In L124-140

14.Response to comment: L 129:Rephrase as : and they were then informed about the EOL protocol.

Responds: After reading the article carefully, we have rewritten the sentence: They were then informed and familiarized with the EOL and HS protocols, CMJ jumps and 30m sprint tests. In L173-175.

15.Response to comment: L 130-131:Please provide additional explanations about the expected effects of the familiarization procedure for experienced subjects.

Responds: Thank you for your valuable suggestions. After carefully reading the reviewers' comments, we have added the following clarification: Participants in this trial typically performed resistance training in the strength and conditioning laboratory once a week as a training task, and therefore had a priori knowledge of the training protocol and testing methods. An additional familiarization exercise was provided prior to the experiment. In L158-160.

16.Response to comment: L 133-139:The procedure implemented to assess CMJ and sprint performance requires further supporting elements. Could the authors indicate for instance why they did not alternate the testing between CMJ and sprint?

Responds: We are grateful to the reviewers for their comments. We are currently hypothesizing in accordance with your opinion that participants did not alternate CMJ and sprint tests during the first 3 sessions based on the hypothesis that cross-testing after one training session (particularly in the EOL protocol) may induce some muscle damage that could lead to biased maximal performance at CMJ or sprint. We also tried to further improve the reliability of our conclusions by conducting the same repeated trials based on the same group of investigators at different time periods. In L144-147.

17.Response to comment:. L 139:Please add the time-delay allowed between the two sequences of this protocol.

Responds: We are very grateful for the reviewers' comments. We would like to express the meaning: Training break of 1 week (14 days) between the two programs. In L141-144.

18.Response to comment:Also, justify that the 1-week washout period is sufficient to alleviate the effects of the first protocol on the testing and effectiveness of the second sequence. (which seems unlikely).

Responds: We greatly appreciate the reviewers' comments. We are currently revising in accordance with your comments to: The authors interrupted the training sessions for 1 week as a washout period (14 days between training sessions) to mitigate the effects of the first protocol on the second protocol tested further research was required. In L141-144.

19.Response to comment:L 149:I recommend the authors to avoid the excessive use of decimals and report the appropriate precision provided by the measurements they performed. 

Responds: We greatly appreciate the reviewers' comments. We have added the table about the basic information of the participates. (age, height, weight, etc.) to the attachment. L 260-265

20.Response to comment:L 161:Should be : "subjects were asked to"

Responds: Thank you for your valuable comments. We have revised the presentation. All subjects were informed about the potential risks and benefits of the current procedures and were asked to signed an informed consent form. In L162.

21.Response to comment:L 170:Please detail what are those "basic information"

Responds: We are very grateful for the reviewers' comments. We have added the table about the basic information of the participates. (age, height, weight, etc.) to the attachment. L 260-265

22.Response to comment:. L 178-180:Do the authors control for the reproducibility between the two trials to ensure the validity of the calculated mean? If so, please provide the accepted range between trials to ensure two maximal performance.

Responds: We greatly appreciate the reviewers' comments. We have added the motion control method between the two trials: All test and training interventions are set up with the appropriate movement control mode. Technical feedback from the field is provided by experienced investigators. Tests and training sessions are re-executed once the subject's movements are incorrectly executed or not at maximum effort. In L178-182.

23.Response to comment:L 181: What was the recovery modality allowed to the subjects between the different time points?

Responds: We are grateful for the reviewers' comments. We added the recovery method for different time points and the participants used passive recovery between time points. In L191-192.

24.Response to comment: L 191-193:Please provide the height of the photoelectric cells.

Responds: We greatly appreciate the reviewers' comments. We have added the height of the equipment: These photocells were mounted 1 m above floor level. In L207.

25.Response to comment:L 205:In accordance with one of my previous comment regarding the lack of details about the "basic information" recorded by means of body scale, the authors should provide additional information regarding the method used to calculate body surface area.

Responds: Thank you for your valuable comments. We appreciate the reviewers' comments. We have added the basic information to the attachment. We are very sorry that our Inbody370 device was not able to provide body scale calculations.

26.Response to comment: L 207-209:Please precise whether the pace was controlled during the downward phase of the half squat. If so, this information should be added to provide an estimate of the braking resistance applied by the subjects.

Responds: Thank you for your valuable comments. We have added the motion control method. Subjects started in a half-squat position and were asked to execute the concentric phase as quickly as possible and to control the braking phase until the knees were flexed to approximately 90 degrees. In L219-222.

27.Response to comment: L 214-215:It seems that the amount of repetitions per set is lacking here. Please amend.

Responds: Thank you for your valuable comments. We have modified the number of reps. For performing 5 sets × 3 reps with a 2-minute rest period between sets. In L229.

28.Response to comment: L 225-227:Provide further explanations about the meaning one his one-way ANOVA. Indeed, your protocol implemented two exercises and three modality. How could the authors therefore performed only a one-way ANOVA?

Responds: We apologize for any confusion caused in the article. In our design, different intensity(3 intensities) was implemented each week for three weeks, and repeated measures of outcome indicators were performed after each trial. Given the specificity of the trial, we viewed the three intensities as independent of each other and therefore performed a one-way repeated measures ANOVA in each intensity. We also did a two-way ANOVA. We have uploaded the statistics in the attachment and hope it will help you to solve the confusion of reading. L 241-256

29.Response to comment: L 233-234:Statistical information are lacking in figures (significant different and not only ANOVA F or p values) forcing the readers to back-and-forth between figures and text. It could be therefore suitable to add the calculated 95%CI on the graphs and symbols to indicate statistical differences.

Responds: Thank you for your valuable comments that would enhance the quality of our manuscript. We have tried as much as we can, however, due to the dense graphics, the graphics appear more confusing after adding 95% CI, which seriously affects the clarity of the graphics. For this reason, we weighed again and again and apologized for not adding 95% CI to the graphics. We have uploaded the statistics in the attachment and hope it will help you to solve the confusion of reading.

30.Response to comment: L 240-243:The comparisons between time points and intensity for pre-HS and pre-EOL is unclear here. It is state in the protocol that each session differed from the other by the testing protocol (HS or EOL) or the intensity (low, med and high). Please amend therefore to understand which variables were compared here.

Responds: Thank you for your valuable comments that so enhance the quality of our manuscript, and We have revised the study design of the manuscript, which we hope will contribute to the understanding of the results. In the first phase (weeks 1-6) participants were randomized into two groups (pre-EOL vs. pre-HS) and in the second phase (weeks 8-13) participants crossed over between the two groups (pre-EOL to post-HS, pre-HS to post-EOL). We also have revised flowchart of the experiment. and re-uploaded it as an attachment. In L126-129

31.Response to comment: L 249: Could you please indicate the degree of freedom for the F value of the ANOVA. In addition, I suggest to avoid the appelation of week1, 2 or 3, should rather precise the intensity that is one of the main independent variable.

Responds: Thank you for your valuable comments that so enhance the quality of our manuscript, and We have made revisions and added information to the manuscript in the appropriate places based on the comments.

32.Response to comment: L 252-253:This sentence requires rephrasing to avoid misinterpretation. Under the current version, this sentence would means that the performance at PRE is significantly related to measurements made at POST. However, it seems that this analysis was not performed, and is not of primary importance for this study.

Responds: We apologize for any confusion caused in the article. We used a two-stage crossover design. We have revised the study design of the manuscript, which we hope will contribute to the understanding of the results. In the first phase (weeks 1-6) participants were randomized into two groups (pre-EOL vs. pre-HS) and in the second phase (weeks 8-13) participants crossed over between the two groups (pre-EOL to post-HS, pre-HS to post-EOL). We also have revised flowchart of the experiment. and re-uploaded it as an attachment. 

33.Response to comment: L 263-265:It could be suitable to report the magnitude of the significant changes in the text to allow the reader a better adoption of the findings.

Responds: Thank you for your valuable comments that so enhance the quality of our manuscript, and We have revised the sloppy presentation. We have add the magnitude of the significant changes in the text, and uploaded the statistics in the attachment and hope it will help you to solve the confusion of reading. L 299

34.Response to comment: L 277-278:This third comparison lacks of support here, and should be presented in the statistics section.

Responds: Thank you for your valuable comments that so enhance the quality of our manuscript, and We have made additional revision in the statistics section. In L248

35.Response to comment:: L 282-283:This sentence is misconducting here since the mention of rest interval suggest that post-measurements would be presented, while it seems that comparison is performed on PRE measurements. Please amend and/ or correct.

Responds: We apologize for any confusion caused in the article. Thank you for your valuable comments that so enhance the quality of our manuscript, and We have modified the experimental design. Since the previous presentation of the experimental design caused confusion in your reading of the results. Again, we apologize. In L124-140

36.Response to comment: L 284-285:Please state clearly this difference in the text

Responds: Thank you for your valuable comments that so enhance the quality of our manuscript, and we have completed the changes in the manuscript. We have added relevant information. In L321-323

37.Response to comment:L 288-290:Again this sentence is misconducting, since is could be understand that the PRE-HS value is significantly correlated to PPO. Please rephrase.

Responds: We apologize for any confusion caused in the article. Thank you for your valuable comments that so enhance the quality of our manuscript, and We have modified the experimental design. Since the previous presentation of the experimental design caused confusion in your reading of the results. we have re-uploaded the flowchart of the experiment and the statistical results. Again, we apologize. In L124-140

38.Response to comment: L 333-335:How could the authors state that the sprint speed of PRE-EOL could be lower than PRE-HS at 9-min rest interval? Once again this sentence is conflicting here and needs to be rephrase.

Responds: We apologize for any confusion caused in the article. Thank you for your valuable comments that so enhance the quality of our manuscript, and We have modified the experimental design. Since the previous presentation of the experimental design caused confusion in your reading of the results. we have re-uploaded the flowchart of the experiment and the statistical results. Again, we apologize. In L124-140

39.Response to comment:L 342-344: Interactions rather revealed significant differences than influenced results. Please rephrase for a better understanding.

Responds: Thank you for your valuable comments. We have revised the inappropriate statement. Change affect to influence. L380-381

40.Response to comment: L 361-362:It would be suitable to differentiate between stretch shortening cycle occurring during running or CMJ for instance and the succession of eccentric and concentric phase with no (or only a few) storage of passive elastic energy (e.g. squatting). 

Responds: We are very grateful for the reviewers' comments. We have added your summary. In L411-413

41.Response to comment:L 369-370:This aim differ from the scientific question addressed in the introduction section, and is therefore not supported by the rational. Please ensure consistency about the objectives of this study throughout the manuscript.

Responds: We are very grateful for the reviewers' comments. We have removed the inappropriate expressions. And revised the presentation in the introduction and the discussion section so that the discussion is consistent with the introduction.

42.Response to comment:L 398-404:There are contradictory findings mentioned here between improvement in CMJ height and PPO due to EOL training, while non-statistical difference is noted. Please based your statement on the statistical findings and not graphical reading.

Responds: We apologize for any confusion caused in the article. Thank you for your valuable comments that so enhance the quality of our manuscript, and we have revised the presentation in the introduction and the discussion section and modified the experimental design. Since the previous presentation of the experimental design caused confusion in your reading of the results. we have re-uploaded the flowchart of the experiment and the statistical results. Again, we apologize. In L124-140

43.Response to comment:L 428-430:This section is quite descriptive and lacks of elements to explain the absence of significant improvement in this study. 

Responds: We are very grateful for the reviewers' comments. We have revised the inappropriate statement. In L430-438

44.Response to comment:L 438-440:I acknowledge that jumping performance is important for basketball players. However, as it is mentioned previously, the effect of PAPE is only transitory and would not be effective during an entire match. 

Responds: We are very grateful for the reviewers' comments. We have revised the description and added a Practical Application section. In L470-477

45.Response to comment: L 447-449:This sentence contradicts the first one of the conclusion section. Please ensure coherence about the presence or absence of PAPE phenomenom in the present study. This question remains unclear while reading this section.

Responds: We are very grateful for the reviewers' comments. We have rewritten the conclusion. In L465-468. We apologize for any confusion caused in the article. Thank you for your valuable comments that so enhance the quality of our manuscript, and We have modified the experimental design. Since the previous presentation of the experimental design caused confusion in your reading of the results. we have re-uploaded the flowchart of the experiment and the statistical results. Again, we apologize. In L124-140

46.Response to comment: L 449-452:Both squatting and flywheel are based on eccentric-concentric contractions here. It appear therefore obvious to precise more specifically the different exercise modalities (e.g.; phase duratin, intensity, ...).

Responds: We are very grateful for the reviewers' comments. We have rewritten the conclusion. In L465-468. We apologize for any confusion caused in the article. Thank you for your valuable comments that so enhance the quality of our manuscript, and We have modified the experimental design. Since the previous presentation of the experimental design caused confusion in your reading of the results. we have re-uploaded the flowchart of the experiment and the statistical results. Again, we apologize. In L124-140

Response to Reviewer #2:

1.Response to comment: Line 46-47:Not clear, rephrase and explain clearly the design.

Responds: We apologize for any confusion caused in the article. Thank you for your valuable comments that so enhance the quality of our manuscript, and We have modified the Methods L 42-51. We have modified the experimental design. Since the previous presentation of the experimental design caused confusion in your reading of the results. we have re-uploaded the flowchart of the experiment and the statistical results. Again, we apologize. In L124-140

2.Response to comment: Lines 49-51: Not useful in an abstract. 

Responds: Thank you for your valuable input. We have removed the inappropriate statement.

3.Response to comment: Lines 52-58:There is no consistency between the methods and the results.

Responds:We apologize for any confusion caused in the article. Thank you for your valuable comments that so enhance the quality of our manuscript, and We have modified the result. L52-66 We have modified the experimental design. Since the previous presentation of the experimental design caused confusion in your reading of the results. we have re-uploaded the flowchart of the experiment and the statistical results. Again, we apologize. In L124-140

4.Response to comment: The last sentence makes no sense since every movement is a combination of concentric/eccentric actions.

Responds: Thank you for your valuable input. We have removed the inappropriate statement.

5.Response to comment: Lines 103-106:It is not clear what “long season” means here. Please explain.

Responds: We are very sorry that our presentation has caused you confusion. We intended to say over a season. In L117

6.Response to comment:Overall, why half-squat and not other squat variations? Please elaborate.

Responds: We are very sorry that our presentation has caused you confusion. We have modified the expression, the flywheel exercise is a half-squat movement pattern, in order to compare the effect of the two programs, the traditional squat also uses the barbell half-squat pattern.

7.Response to comment:: Line 195:PAP or PAPE?

Responds: We are very grateful for the reviewers' comments. For the reviewers' valuable comments, we add the study by Blazevich and Babault. In this study, the term PAPE was used in relation to the enhancement of muscle contraction after conditioning activity. In L80-81

8.Response to comment:The statistical analysis should be rerun: it is not clear to me why two different two-way analysis were performed instead of a three-way.

Responds: Thank you for such valuable comments, and we have asked several statistical experts to conduct a thorough evaluation of the statistical methods in the manuscript, and they believe that we have a cross-sectional design, however a repeated measures design nested within it that measures outcome indicators at different time points, so the two-factor ANOVA and single-cause repeated measures ANOVA used in the current manuscript support our conclusions well.

9.Response to comment:: This is a major point that should be addressed. Therefore, I have stopped my review here, since the results should be rewritten accordingly.

Responds: For this, I am very sorry. We have made many revision changes to address the issues you have raised. Please continue to review our manuscript. Thank you.

Response to Reviewer #3:

1.Response to comment: The manuscript refers to post-activation potentiation (PAP) throughout – attributed to phosphorylation of myosin regulatory light chains. 

Responds: We are very grateful for the reviewers' comments. After carefully reading the reviewers' comments, We adopted the concept of PAPE, complemented the relevant literature, and described the mechanisms associated with PAP and PAPE. In L83-90.

2.Response to comment: The terms used to describe the exercises are quite vague. For example, an eccentric overload could be achieved in many different ways and a ‘loaded half squat’ could also be achieved in different ways. In fact, it could be argued that both exercises are ‘loaded half squats.

Responds: We are very grateful for the reviewers' comments. We have carefully reviewed and rewritten the methods. We compare flywheel eccentric overloading and barbell squatting.

3.Response to comment: The loads used for the barbell squat are individual-specific (a percentage of 1RM) but the flywheel inertias are not. This warrants further consideration or discussion.

Responds: We are very grateful for the reviewers' comments. We acknowledge that comparing the effects of two different movement patterns is a great challenge, and we have discussed it. In L402-410

4.Response to comment: In the statistical analysis section, it is stated that one-way repeated measures ANOVAs were performed. However, a few lines later on talk about interaction effects, which suggest that at least a two-way ANOVA was performed. Responds: Thanks to such valuable comments, we have rephrased the study design and uploaded the study roadmap with the results of data analysis. We have also considered carefully and invited statistical experts for a thorough review of the study methodology in the manuscript. However, they felt that they believed that we were a cross-sectional design and, with a repeated measures design nested within it that measured outcome indicators at different time points, and therefore, suggested that a two-factor ANOVA and a single-cause repeated measures ANOVA would support our conclusions well in the manuscript.

Although we did not make the assumption that the two minimum strengths (flywheel and barbell) are equal, we standardized the movement pattern of the half squat (flywheel and barbell) and We used the lightest weight and inertia torque of the two devices as the low intensity for this test. We also consider this a great challenge.

5.Response to comment: Specific comments:Authors: It is not clear how there can be three co-first authors.

Responds: We have revised to two co-first authors.

6.Response to comment: Short title: Compared to what?

Responds: We are very grateful for the reviewers' comments. Flywheel half-squat compare to barbell half-squat exercise.

7.Response to comment: Data availability statement: I suggest that you upload the data alongside the manuscript for readers and reviewers in line with journal policy.

Responds: We are very grateful for the reviewers' comments. We have uploaded the statistics rusult in the attachment.

8.Response to comment: Line 52:PPO should be defined. Peak power output? It would also be useful to include some statistical results within the abstract.

Responds: We greatly appreciate the reviewers' comments. We have changed the definition of PPO to Peak power output (PPO). In L52

9. Response to comment: Lines 74-75:This statement is too dramatic/ambitious and should be reduced in everity.

Responds: We greatly appreciate the reviewers' comments. We have modified the inappropriate statement. In L80-93

10. Response to comment:Line 95:Other studies have investigated the relationship between flywheel moment of inertia and velocity or power (concentric and eccentric) during flywheel half squats – see https://doi.org/10.1080/02640414.2020.1860472

Responds: We greatly appreciate the reviewers' comments. We have added and revised the presentation in the preface and discussion sections, and added references.

11. Response to comment: Lines 101-106:Although not on jumping sports, quite a few PAPE studies have focused on vertical or horizontal jumps. 

Responds: We are very grateful for the reviewers' comments. As the presentation has been rewritten, we have added relevant references in different places.

12. Response to comment: Lines 118-120:It’s unclear how many sessions this is in total. How many familiarisation sessions were performed if any? 

Responds: We are very grateful for the reviewers' comments. 12 sessions in total, an additional 1 familiarization sessions were performed, and we have cited references and modified the presentation accordingly. The corresponding statement was modified in the method section. We have modified the experimental design. Since the previous presentation of the experimental design caused confusion in your reading of the results. we have re-uploaded the flowchart of the experiment and the statistical results. Again, we apologize. In L124-140

13. Response to comment: Line 148:Why 12 participants? This number should be justified. See https://doi.org/10.1525/collabra.33267

Responds: Thank you for your valuable comments that so enhance the quality of our manuscript, and we have the changes in the manuscript. The entire membership of a basketball team participated in the trial, originally 18, with 6 withdrawing due to injury. In L260-265

14. Response to comment:Line 161 and elsewhere:I suggest being consistent with ‘participants’ or ‘subjects’ – the section is called ‘Participants’ so it would be good to stick with this.

Responds: Modified to: participants.

15. Response to comment: Lines 178 and 188:Why were two trials averaged? Why two and why an average?

Responds: We are very grateful for the reviewers' comments. We have revised the presentation. Two tests were conducted to ensure that it was a maximum effort and that the results met the criteria of the movement pattern. The best performance was used for statistical analysis. In L188 and L202

16. Response to comment: Lines 185-186:How were these parameters calculated? There are multiple possible methods for some of them.

Responds: Thank you for your valuable comments. The jump performance parameters were measured by Smart Jump (Fusion Sport; Australia) and standardized to the CMJ movement pattern. In L193-195

17. Response to comment: Line 192:What was the starting position for the sprint? In line with the timing gate or slightly behind?

Responds: We are very grateful for the reviewers' comments. We have revised the presentation. In L207-209

18. Response to comment:Line 201:Why 3 sets x 6 reps? The volume PAPE paper mentioned earlier could be used here, but it is important to justify these choices.

Responds: Thank you for your valuable comments that so enhance the quality of our manuscript. The exercise was performed by reference: 10.1519/JSC.0000000000003005. The applicability of the method will be studied in depth subsequently. Maroto-Izquierdo et al. (10.5114/biolsport.2020.96318) found that a single set (i.e., 6 repetitions) of the half-squat flywheel exercise performed with the optimal concentric peak power intensity in physically active young men led to significant acute gains in CMJ jump height.

19 Response to comment:Lines 208-209: Were participants instructed to resist throughout the eccentric phase or only in a certain portion of it?

Responds: Thank you for your valuable comments that so enhance the quality of our manuscript, and we have revised in the manuscript. Eccentric process is resisted, knee flexion control at 90 degrees as the limit for centripetal contraction. In L219-224

20. Response to comment:Lines 232-233:These should be reworded with signs such as ‘greater than or equal to’ to ensure that there are no gaps. For example, 1.195 and 1.995 currently have no category.

Responds: Thank you for your valuable comments. We have rephrased trivial (<0.200), small (0.200 - 0.599), moderate (0.600 - 1.199) and large (1.200 -1.999), very large (≥2.000). In L254-255

21. Response to comment:I suggest adding tables to make the results clearer.

Responds: Thank you for your valuable comments. We have added the statistical result in the attachment.

22. Response to comment:Line 368:‘shown’ may be better than ‘proved’ – to show more uncertainty.

Responds: Thank you for your valuable comments. We have revised the inappropriate statement.

23. Response to comment: It would be good to re-summarise the main overall results early in the Discussion.

Responds: Thank you for your valuable comments. We have rewritten the main overall results early in the Discussion. In L397-401

24. Response to comment:Lines 369-370: More justification is needed for why basketballers might be different to other populations and why results might not continue over a season.

Responds: Thank you for your valuable comments. We have added Practical Applications. In L 470-477

25. Response to comment: Line 382:What is meant by ‘ground-lifting’?

Responds: Thank you for your valuable comments. Due to inaccurate translation, we have removed the expression.

26. Response to comment: Lines 384-387:Some of the studies I mentioned above (e.g. the effects of inertia on PAPE) may be useful here. Likewise, for lines 390-391 where the differential effects of peak power / velocity / force are discussed.

Responds: Thank you for your valuable comments. We have revised the presentation and cited the relevant literature. In L 418-423

27. Response to comment: Line 397: Could this be controlled for within the statistical analysis?

Responds: Thank you for your valuable comments. We did make relevant controls, re-upload the study roadmap and data statistics, and rewrote the study design to make it easier to understand.

28. Response to comment:Line 411:Was this ‘tendency’ significant? If not then it should not be discussed as an effect. The same applies in line 448 (if not then this should not be part of the conclusion).

Responds: Thank you for your valuable comments. We have revised the word to effect in the relevant position.

29. Response to comment: Lines 418/420/421-consistency needed around minutes / -minutes / min

Responds: Thank you for your valuable input. For consistency of presentation, we have modified the expression to read: min.

We gratefully appreciate for your valuable suggestion

We tried our best to improve the manuscript and made some changes in the manuscript. These changes will not influence the content and framework of the paper. And all the changes are marked in red in revised paper.

We appreciate for Editors/Reviewers’ warm work earnestly, and hope that the correction will meet with approval.

Once again, thank you very much for your comments and suggestions.

Yours sincerely,

Jian Sun, School of Athletic Training, Guangzhou Sport University, Guangzhou, P.R. China, E-mail: sunjian@gzsport.edu.cn . Tel:+8613728059899

---

## [Decision Letter · Decision Letter 1]

20 Jul 2022

PONE-D-22-02783R1Flywheel eccentric overload exercises versus barbell half squats for basketball players: which is better for induction of post-activation potentiation enhancement?PLOS ONE

Dear Dr. Sun,

Thank you for submitting your manuscript to PLOS ONE. After careful consideration, we feel that it has merit but does not fully meet PLOS ONE’s publication criteria as it currently stands. Therefore, we invite you to submit a revised version of the manuscript that addresses the points raised during the review process. Despite significant improvements, methodological clarifications need to be added to the current manuscript, as well as approriate answer to reviewers' comments.

We look forward to receiving your revised manuscript.

Kind regards,

Laurent Mourot

Section Editor

PLOS ONE

Reviewers' comments:

Reviewer's Responses to Questions

**Comments to the Author**

1. If the authors have adequately addressed your comments raised in a previous round of review and you feel that this manuscript is now acceptable for publication, you may indicate that here to bypass the “Comments to the Author” section, enter your conflict of interest statement in the “Confidential to Editor” section, and submit your "Accept" recommendation.

Reviewer #1: (No Response)

Reviewer #3: (No Response)

2. Is the manuscript technically sound, and do the data support the conclusions?

Reviewer #1: Yes

Reviewer #3: Partly

3. Has the statistical analysis been performed appropriately and rigorously? 

Reviewer #1: Yes

Reviewer #3: No

4. Have the authors made all data underlying the findings in their manuscript fully available?

Reviewer #1: Yes

Reviewer #3: No

5. Is the manuscript presented in an intelligible fashion and written in standard English?

Reviewer #1: Yes

Reviewer #3: Yes

6. Review Comments to the Author

Reviewer #1: I thank the authors for considering my comments that enable significant improvements in the manuscript. It remains however some misunderstandings from my first revision that should be corrected before publication. Please find below these elements :

Abstract

L 58-59, 60: My previous comment about the magnitude of PPO increase considered the value of the increase (in % pre for instance) rather than the effect size of the statistical analysis. This would better inform the readers about the main findings of this study in the abstract.

L 145-148: I understand the statement of the authors considering the greater risk to induce muscle impairments with a 30-m maximal sprint. However, the repeated bout effect associated to eccentric strength training impact the magnitude and etiology of the experimental training’s residual effects. Therefore, it is conceivable that the PAPE effects observed for sprint performance after 3 days of training will not represent the effects that would be observed after the first occurrence to eccentric strength training. I suggest the authors to add this limit in the present article.

L 171: “basic information” is useless and could be remove to avoid excessive wordings (e.g. “participants’ height, age, weigth,…” is satisfactory).

Results:

L296-299: it seems that my comment in R1 was not well understood by the authors, which contribute also to confusion in the abstract. By “amplitude” I would say, could the authors provide the percentage increase (or decrease) in participants’ performance? The effect size refers to the power of the statistical analysis, but that is of moderate importance for practical application of the protocol from coaches and strength conditioners. Please add here, and in the abstract, the % change from PRE for your values.

L 380-381: again my comment in revision #1 was not understand. The statistical results of the ANOVA describe the differences observed in your study. This analysis (and the interaction provided here) made a posteriori could not “affect” or “influence” the sprint speed. The results of the ANOVA showed differences between the post-EOL and post-HS groups. Please correct carefully.

Reviewer #3: General comments:

I previously stated that, among other issues, it was unclear what statistical analyses had been performed. The previously recommended revisions have mostly been addressed/clarified, although a few areas remain difficult to interpret.

Comment 1: I still do not believe it is appropriate to consider low/medium/high intensity as being equivalent between the two exercises. For example, what rationale is there to assume that 40% and 0.015 are both low and that 80% and 0.075 are both high.? It is also worth noting that 80% is double 40% but 0.075 is five times 0.015. This may affect the statistical analysis used, but if not then it should at least be discussed as an assumption upon which the results rest.

Comment 2: In the statistical analysis section, it is much improved but still could be clearer exactly what variables are being assessed in what combinations/conditions. I suggest taking more words to spell this out in great detail. For example, the one-way ANOVA on line 247-248 can’t be the 2x2 pre/post HS/EOL and also can’t be the 2x1 pre/post or HS/EOL, so it’s not clear where the one-way ANOVA comes from. Generally, the statistics are easier to follow in the Results section because each individual result is discussed one at a time.

Comment 3: Despite changing to ‘barbell half squat’ and ‘flywheel eccentric overload’, the abbreviations HS and EOL are used. As previously stated, both are half squats and there are many ways of eliciting an eccentric overload. I suggest simply using the terms ‘barbell’ and ‘flywheel’ or including B and F in the abbreviations if necessary.

Comment 4: I also suggest avoiding the terms pre and post as they do not refer to the pre and post PAPE effect, which is confusing.

Comment 5: When referring to peak power or impulse, in many cases it is not stated whether this is during a jump or a sprint. This should always be clear in case a reader only reads specific sections.

Specific comments:

Title: This should be ‘performance’, nit ‘potentiation’ for PAPE

Lines 23-24: The authors should be commended for making the statistical results available in full, although this should not be stated as making the underlying data available.

Line 59: The ‘stages’ have not been explained yet so this is hard for someone to follow if only reading the abstract. These stages are called phases elsewhere, so it should be consistent.

Line 198: What method is used to calculate these? I know it is from Smart Jump, but readers should be told what method is used for the calculations.

Lines 254-255: There are still gaps – e.g. 0.5995. I suggest e.g. 0.2 ≤ small < 0.6; 0.6 ≤ moderate < 1.2.

Lines 260-265: this information should be in the participants section of the Methods.

Line 267: No criteria were given in the Methods for determining moderate or excellent reliability.

Line 276: Please do not say ;tended to increase’ if it was not significant.

7. PLOS authors have the option to publish the peer review history of their article (what does this mean?). If published, this will include your full peer review and any attached files.

Reviewer #1: No

Reviewer #3: No

---

## [Author Response · Author response to Decision Letter 1]

21 Aug 2022

Question#1: Abstract: L 58-59, 60: My previous comment about the magnitude of PPO increase considered the value of the increase (in % pre for instance) rather than the effect size of the statistical analysis. This would better inform the readers about the main findings of this study in the abstract.

Response: Thank you very much for your suggestion. We have replaced effect size with percentages. Please see L58-59 and L61.

Question#2: L 145-148: I understand the statement of the authors considering the greater risk to induce muscle impairments with a 30-m maximal sprint. However, the repeated bout effect associated to eccentric strength training impact the magnitude and etiology of the experimental training’s residual effects. Therefore, it is conceivable that the PAPE effects observed for sprint performance after 3 days of training will not represent the effects that would be observed after the first occurrence to eccentric strength training. I suggest the authors to add this limit in the present article.

Responds: Thank you very much for your suggestion. We have added the relevant content. Please see L482-487.

Question#3: L 171: “basic information” is useless and could be remove to avoid excessive wordings (e.g. “participants’ height, age, weigth,…” is satisfactory).

Responds: Thank you very much for your suggestion. We have removed this section.

Question#4: Results: L296-299: it seems that my comment in R1 was not well understood by the authors, which contribute also to confusion in the abstract. By “amplitude” I would say, could the authors provide the percentage increase (or decrease) in participants’ performance? The effect size refers to the power of the statistical analysis, but that is of moderate importance for practical application of the protocol from coaches and strength conditioners. Please add here, and in the abstract, the % change from PRE for your values.

Responds: Thank you very much for your suggestion. We have added the corresponding percentages. Please see L322-323, L360, L395, L401-402, L408.

Question#5: L 380-381: again my comment in revision #1 was not understand. The statistical results of the ANOVA describe the differences observed in your study. This analysis (and the interaction provided here) made a posteriori could not “affect” or “influence” the sprint speed. The results of the ANOVA showed differences between the post-EOL and post-HS groups. Please correct carefully.

Responds: Thank you very much for your suggestion. We have corrected the relevant content. Please see L354-356, L403-406.

Reply to the third reviewer as follows:

Question#6: I still do not believe it is appropriate to consider low/medium/high intensity as being equivalent between the two exercises. For example, what rationale is there to assume that 40% and 0.015 are both low and that 80% and 0.075 are both high.? It is also worth noting that 80% is double 40% but 0.075 is five times 0.015. This may affect the statistical analysis used, but if not then it should at least be discussed as an assumption upon which the results rest.

Response: Thank you for your suggestion. We're not looking at low/medium/high intensity as an equivalence between the two sports. The low, medium, and high intensity of both types of exercise are relative terms. For the choice of training intensity in BHS, we refer to the study of Buscà et al. [1]. For the choice of training intensity in FEOL, we refer to the study of Petersen et al. [2].

[1] B. Buscà, J. Aguilera-Castells, J. Arboix-Alió, A. Miró, A. Fort-Vanmeerhaeghe, J. Peña, Influence of the Amount of Instability on the Leg Muscle Activity During a Loaded Free Barbell Half-Squat, International journal of environmental research and public health, 17 (2020).

[2] J. Petersen, K. Thorborg, M.B. Nielsen, E. Budtz-Jørgensen, P. Hölmich, Preventive effect of eccentric training on acute hamstring injuries in men's soccer: a cluster-randomized controlled trial, The American journal of sports medicine, 39 (2011) 2296-2303.

Question#7: In the statistical analysis section, it is much improved but still could be clearer exactly what variables are being assessed in what combinations/conditions. I suggest taking more words to spell this out in great detail. For example, the one-way ANOVA on line 247-248 can’t be the 2x2 pre/post HS/EOL and also can’t be the 2x1 pre/post or HS/EOL, so it’s not clear where the one-way ANOVA comes from. Generally, the statistics are easier to follow in the Results section because each individual result is discussed one at a time.

Response: Thank you for your suggestion. The main method of our study was a crossover design, however, to observe time effects, we further added a repeated measures design. Finally, to analyze the intervention mode and time effects, we used a combination of one-way ANOVA and repeated measures ANOVA. We have supplemented the relevant statement in the Methods section. Please see L266-275.

Question#8: Despite changing to ‘barbell half squat’ and ‘flywheel eccentric overload’, the abbreviations HS and EOL are used. As previously stated, both are half squats and there are many ways of eliciting an eccentric overload. I suggest simply using the terms ‘barbell’ and ‘flywheel’ or including B and F in the abbreviations if necessary.

Responds: Thank you for your suggestion. We have made changes to the relevant content. Please see Revised Manuscript and Revised Figures.

Question#9: I also suggest avoiding the terms pre and post as they do not refer to the pre and post PAPE effect, which is confusing.

Responds: We are sorry for our inappropriate presentation. We have replaced "pre" and "post" with "I" and "II", which represent the two training stages, Stage I and Stage II, respectively. We have modified the relevant content in the text and figures. Please see Revised Manuscript and Revised Figures.

Question#10: When referring to peak power or impulse, in many cases it is not stated whether this is during a jump or a sprint. This should always be clear in case a reader only reads specific sections.

Responds: We greatly appreciate the reviewer's comments. For the reviewer's valuable comments, we have stated in the text that it is peak power or impulse during the jump. Please see L207, L339, L368.

Question#11: Title: This should be ‘performance’, nit ‘potentiation’ for PAPE.

Responds: We are sorry for our carelessness. We have changed "potentiation" to "performance". Please see the Title and L72.

Question#12: Lines 23-24: The authors should be commended for making the statistical results available in full, although this should not be stated as making the underlying data available.

Responds: We have made changes to the Data Availability Statement. Please see L24.

Question#13: Line 59: The ‘stages’ have not been explained yet so this is hard for someone to follow if only reading the abstract. These stages are called phases elsewhere, so it should be consistent.

Responds: We are sorry for our inappropriate presentation. We have supplemented the relevant explanation of "stages" in the Abstract. In addition, we also corrected the incomprehensible “Phases” in the main text to the corresponding “stages-I” and “stage-II”. Please see L60, and Revised Manuscript.

Question#14: Line 198: What method is used to calculate these? I know it is from Smart Jump, but readers should be told what method is used for the calculations.

Responds: Your comments are greatly appreciated, and we have added the formula in the manuscript. Please see L208-212.

Question#15: Lines 254-255: There are still gaps – e.g. 0.5995. I suggest e.g. 0.2 ≤ small < 0.6; 0.6 ≤ moderate < 1.2.

Responds: Thank you for your suggestion. We have made changes based on comments. Please see L269-270.

Question#16: Lines 260-265: this information should be in the participants section of the Methods.

Responds: We are very grateful to the reviewers for their suggestions. We have added the information from this section to the Methods section. Please see L169-172.

Question#17: Line 267: No criteria were given in the Methods for determining moderate or excellent reliability.

Responds: We refer to the definition criteria of moderate and excellent reliability in the studies of Koo et al [1] and Cormack et al [2]. Briefly, ICC values less than 0.5 are indicative of poor reliability, values between 0.5 and 0.75 indicate moderate reliability, values between 0.75 and 0.9 indicate good reliability, and values greater than 0.90 indicate excellent reliability. A CV of ≤10% was set as the criterion to declare a variable as reliable. A 10% CV cut-off may encourage the examination of variables other than those possessing the highest reliability in future research. We added the relevant content in Methods. Please see L262-266.

[1] T.K. Koo, M.Y. Li, A Guideline of Selecting and Reporting Intraclass Correlation Coefficients for Reliability Research, Journal of chiropractic medicine, 15 (2016) 155-163.

[2] S.J. Cormack, R.U. Newton, M.R. McGuigan, T.L. Doyle, Reliability of measures obtained during single and repeated countermovement jumps, International journal of sports physiology and performance, 3 (2008) 131-144.

Question#18: Line 276: Please do not say ;tended to increase’ if it was not significant.

Responds: Thank you for your suggestion. We have corrected the inappropriate representation. Please see L299.

---

## [Decision Letter · Decision Letter 2]

14 Sep 2022

PONE-D-22-02783R2Flywheel eccentric overload exercises versus barbell half squats for basketball players: which is better for induction of post-activation performance enhancement?PLOS ONE

Dear Dr. Sun,

Thank you for submitting your manuscript to PLOS ONE. After careful consideration, we feel that it has merit but does not fully meet PLOS ONE’s publication criteria as it currently stands. Therefore, we invite you to submit a revised version of the manuscript that addresses all the points raised during the review process. Please submit your revised manuscript by Oct 29 2022 11:59PM. If you will need more time than this to complete your revisions, please reply to this message or contact the journal office at plosone@plos.org. Please include the following items when submitting your revised manuscript:A rebuttal letter that responds to each point raised by the academic editor and reviewer(s). You should upload this letter as a separate file labeled 'Response to Reviewers'.A marked-up copy of your manuscript that highlights changes made to the original version. You should upload this as a separate file labeled 'Revised Manuscript with Track Changes'.An unmarked version of your revised paper without tracked changes. You should upload this as a separate file labeled 'Manuscript'.

We look forward to receiving your revised manuscript.

Kind regards,

Laurent Mourot

Section Editor

PLOS ONE

Reviewers' comments:

Reviewer's Responses to Questions

**Comments to the Author**

1. If the authors have adequately addressed your comments raised in a previous round of review and you feel that this manuscript is now acceptable for publication, you may indicate that here to bypass the “Comments to the Author” section, enter your conflict of interest statement in the “Confidential to Editor” section, and submit your "Accept" recommendation.

Reviewer #1: (No Response)

Reviewer #3: (No Response)

2. Is the manuscript technically sound, and do the data support the conclusions?

Reviewer #1: Partly

Reviewer #3: Yes

3. Has the statistical analysis been performed appropriately and rigorously? 

Reviewer #1: No

Reviewer #3: Yes

4. Have the authors made all data underlying the findings in their manuscript fully available?

Reviewer #1: No

Reviewer #3: Yes

5. Is the manuscript presented in an intelligible fashion and written in standard English?

Reviewer #1: Yes

Reviewer #3: Yes

6. Review Comments to the Author

Reviewer #1: Abstract:

L 53: I suggest the authors to precise what time points are considered here? Is it only immediately after exercise, or at all time points? The sentence is misconducting.

L 57-59: was there a difference between FEOL and BHS at baseline? Please amend.

L60-62: the use of stage I and II is unclear here, and does not refer to one of the objectives stated previously. I suggest to remove this information in the abstract to make it clearer.

L 64-65: this sentence is unclear. If no change on sprint speed was observed after the two training protocols, therefore there was no effect. Please rephrase to make this sentence less confuse.

L 70: would you mean FEOL training?

Introduction

L 79: I suggest to consider rephrasing into “enhancement in explosive sport performance”

L 93: replace for “contractility” or “contractile function” depending on the meaning of your sentence

Methods

L 130: “and in stage-II, they crossed over…”

L 135: “were set before training, …”

L 169: it is unclear whether the 6 players dropped out because of injuries occurring during the experimental protocol, or were not included because of the inclusion criteria (line 158). Please be consistent throughout the paragraph.

L 179: as mentioned in a previous revision, the InBody 370 is a body composition analyzer, and could therefore not measure participant’s height. Please indicate the scale used to measure this variable.

L 198: words are lacking here, following what?

L 224: “On the GO signal” could be better

Results

The same problem still remains in this third version of the manuscript about interpretation of the statistical analysis. Authors should carefully translate their result, and mentioned difference, increase or decrease when statistics prove it. When no statistical difference is findings, therefore there is no increase or decrease of the variables, even if the numbers are not the same. This misconducting makes the results fallacious and decrease the quality of the manuscript. A real effort should be provided here to describe clearly and decently these findings.

Table 1: “indicates significant difference” or “differ significantly “. Under the present form this sentence is unclear

Discussion

L 422-426: the summary of the main findings is unclear. The first sentence states that no performance improvement was noted for CMJ and sprint after the two protocols. The second sentence argues that FEOL training can increase jump height to a greater extent that BHS. These sentences, under the present form, are therefore contradictory. Would the authors say that jump heights at 3, 6 9 and 12 min were greater for FEOL than BHS? If yes, then rephrase to correspond.

L 434-435: please state clearly how this performance could be impacted? Improvement? Decrease?

L 450: as state previously for results section, in absence of statistical difference, no difference exists. Please correct to correspond.

Reviewer #3: The majority of my previous comments have been addressed. Thank you. I only have two remaining comments:

Comment 1: Thank you for your response regarding the low/medium/high intensity for each exercise. I understand that they are not intended to be equivalent intensities between exercises. However, your use of an ANOVA may dictate that they are considered to be equal. For example, does this statistical model assume that you have 3 intensities (low/medium/high) at each of 2 exercises, and then compare? The main and especially the interaction effects may depend on this assumption, so it is worth stating/discussing. If this test is not performed then it is less of an issue, but perhaps giving the exact intensity in tables and results would be better than referring to two different ‘low’, two different ‘medium’, etc. is they are not equivalent.

Comment 2: Please check your equations in lines 210-212. I get different answers using these compared to standard equations. It is also not clear to me why flight time is divided by 1000 (instead of 2). If it is recorded in ms instead of s then this should be clear. I suggest indicating that the PPO is an estimate and not a measurement. If possible, I also suggest using g as 9.81 instead of 9.8.

7. PLOS authors have the option to publish the peer review history of their article (what does this mean?). If published, this will include your full peer review and any attached files.

Reviewer #1: No

Reviewer #3: No

---

## [Author Response · Author response to Decision Letter 2]

29 Sep 2022

Dear editor and dear reviewers

Regarding the manuscript ID (NO.PONE-D-22-02783) entitled " Eccentric overload exercises versus loaded half squats for basketball players: which is better for induction of postactivation potentiation?". Thank you for your letter and the reviewers' comments on our manuscript entitled " Eccentric overload exercises versus loaded half squats for basketball players: which is better for induction of postactivation potentiation?" (NO.PONE-D-22-02783) were evaluated. These comments were valuable and helpful to us in revising and improving the paper, as well as important guidance for our research. We have carefully studied these comments and made revisions, which we hope will be approved by all of you. The revised parts are marked in red in the thesis. The main corrections in the paper and the responses to the reviewers are listed below.

Reply to the first reviewer as follows:

Question#1: Abstract: L 53: I suggest the authors to precise what time points are considered here? Is it only immediately after exercise, or at all time points? The sentence is misconducting. 

Responds: Thank you very much for your valuable advice. We have made changes to the relevant content.Please see L53.

Question#2: L60-62: the use of stage I and II is unclear here, and does not refer to one of the objectives stated previously. I suggest to remove this information in the abstract to make it clearer.

Responds: Thank you for your valuable advice. We have removed this information from the Abstract.

Question#3: L 64-65: this sentence is unclear. If no change on sprint speed was observed after the two training protocols, therefore there was no effect. Please rephrase to make this sentence less confuse.

Responds: Thank you for your valuable advice. We have made changes to the relevant content.Please see L65-66.

Question#4: L 70: would you mean FEOL training?

Responds: Thanks for the tip. We have made changes to the relevant content.Please see L73.

Question#5: Introduction: L 79: I suggest to consider rephrasing into “enhancement in explosive sport performance” 

Responds: We have made changes to the relevant content.Please see L82.

Question#6: L 93: replace for “contractility” or “contractile function” depending on the meaning of your sentence

Responds: Thank you for your valuable advice. We have made changes to the relevant content. Please see L96.

Question#7: Methods: L 130: “and in stage-II, they crossed over…”

Responds: Thank you for your prompt. We have revised the relevant content. Please see L135-136.

Question#8: L 135: “were set before training, …”

Responds: Thank you for your tip. We have revised the relevant content. Please see L140.

Question#9: L 169: it is unclear whether the 6 players dropped out because of injuries occurring during the experimental protocol, or were not included because of the inclusion criteria (line 158). Please be consistent throughout the paragraph. 

Responds: Thank you very much for your valuable advice. We have revised the relevant content. Please see L164.

Question#10: L 179: as mentioned in a previous revision, the InBody 370 is a body composition analyzer, and could therefore not measure participant’s height. Please indicate the scale used to measure this variable. 

Responds: Thank you very much for your valuable advice. We have revised the relevant content. Please see L184-185.

Question#11: L 198: words are lacking here, following what? 

Responds: Thank you for your suggestion. We have changed the inappropriate expression. Please see L204-206.

Question#12: L 224: “On the GO signal” could be better

Responds: Sorry. We have made changes to the inappropriate statements. Please see L232.

Question#13: Results: The same problem still remains in this third version of the manuscript about interpretation of the statistical analysis. Authors should carefully translate their result, and mentioned difference, increase or decrease when statistics prove it. When no statistical difference is findings, therefore there is no increase or decrease of the variables, even if the numbers are not the same. This misconducting makes the results fallacious and decrease the quality of the manuscript. A real effort should be provided here to describe clearly and decently these findings.

Responds: We have revised it according to your valuable suggestion. Please see L56-60, L324-329, L340-342, L344-349, L359-362, L364-366, L373-375, L390-392, L412-415, L423-425, and L430-431.

Question#14: Table 1: “indicates significant difference” or “differ significantly “. Under the present form this sentence is unclear

Responds: We have revised it according to your valuable suggestion. Please see Table 1.

Question#15: Discussion

L 422-426: the summary of the main findings is unclear. The first sentence states that no performance improvement was noted for CMJ and sprint after the two protocols. The second sentence argues that FEOL training can increase jump height to a greater extent that BHS. These sentences, under the present form, are therefore contradictory. Would the authors say that jump heights at 3, 6 9 and 12 min were greater for FEOL than BHS? If yes, then rephrase to correspond.

Responds: We have revised it according to your valuable suggestion. Please see L444-446.

Question#16: L 434-435: please state clearly how this performance could be impacted? Improvement? Decrease?

Responds: We apologise for our inappropriate statement. This part is speculation on our part and we are not sure whether these factors actually have an impact on sports performance. What we were actually trying to convey was the advantage of the current crossover design. We have amended this. Please see L452-458.

Question#17: L 450: as state previously for results section, in absence of statistical difference, no difference exists. Please correct to correspond.

Responds: Thank you for your valuable advice. We have revised the relevant content. Please see L311, and L470-471.

Reply to the third reviewer as follows:

Question#18: Thank you for your response regarding the low/medium/high intensity for each exercise. I understand that they are not intended to be equivalent intensities between exercises. However, your use of an ANOVA may dictate that they are considered to be equal. For example, does this statistical model assume that you have 3 intensities (low/medium/high) at each of 2 exercises, and then compare? The main and especially the interaction effects may depend on this assumption, so it is worth stating/discussing. If this test is not performed then it is less of an issue, but perhaps giving the exact intensity in tables and results would be better than referring to two different ‘low’, two different ‘medium’, etc. is they are not equivalent.

Responds: Thank you for asking the rigorous questions. At the outset of the design, it was noted that the comparison of the two devices for the semi-squat exercise would be the biggest challenge of this thesis. We have categorised the intensity according to the flywheel rotational inertia attached to the equipment as low (0. 015 kg∙m2), medium (0.035 kg∙m2) and high (0.075 kg∙m2), referencing the Petersen J,et al.[1] literature. In addition, the intensity was adjusted to low (40%), medium (60%) and high (80%) according to the number of sets, repetitions and individual 1RM, Gourgoulis V,et al.[2] ,Fukutani A.[3] ,Buscà B,et al.[4] , and other literature. We agree with you and do assume that they are equal, but in reality they are not. We will follow your suggestion to note the corresponding strengths in numerical form for modification.Please see L63-65, L68-69, L125-126, L142-143, L307-309, L315-317, L319-324, L338-339, L364, L386-389, L394-396, L419-421, L489-491, L521.

[1]Petersen J, Thorborg K, Nielsen MB, Budtz-Jørgensen E, Hölmich P. Preventive effect of eccentric training on acute hamstring injuries in men's soccer: a cluster-randomized controlled trial. Am J Sports Med. 2011 Nov;39(11):2296-303. doi: 10.1177/0363546511419277. Epub 2011 Aug 8. PMID: 21825112.

[2] Gourgoulis V , Aggeloussis N, Kasimatis P, et al. Effect of a submaximal half-squats warm-upprogram on vertical jumping ability[J]. J Strength Cond Res, 2003, 17(2):342-344. doi: 10.1519/1533-4287(2003)017<0342:eoashw>2.0.co;2

[3] Fukutani A, Takei S, Hirata K, et al. Influence of the intensity of squat exercises on thesubsequen jump performance[J]. J Strength Cond Res, 2014, 28(8):2236-2243. doi:10.1519/JSC.0000000000000409.

[4] Buscà B, Aguilera-Castells J, Arboix-Alió J, Miró A, Fort-Vanmeerhaeghe A, Peña J. Influence of the Amount of Instability on the Leg Muscle Activity During a Loaded Free Barbell Half-Squat. Int J Environ Res Public Health. 2020 Oct 31;17(21):8046. doi: 10.3390/ijerph17218046. PMID: 33142906; PMCID: PMC7672597.

Question#19: Please check your equations in lines 210-212. I get different answers using these compared to standard equations. It is also not clear to me why flight time is divided by 1000 (instead of 2). If it is recorded in ms instead of s then this should be clear. I suggest indicating that the PPO is an estimate and not a measurement. If possible, I also suggest using g as 9.81 instead of 9.8. 

Responds: Thank you for your valuable advice. We have specified the unit ms in flight time and have used the PPO as an estimate rather than a measurement as you suggested, which has been very enlightening. Also changed the gravity (g) to 9.81 m/s. Please see L216-219.

We gratefully appreciate for your valuable suggestion

We tried our best to improve the manuscript and made some changes in the manuscript. These changes will not influence the content and framework of the paper. And all the changes are marked in red in revised paper.

We appreciate for Editors/Reviewers’ warm work earnestly, and hope that the correction will meet with approval.

Once again, thank you very much for your comments and suggestions.

Yours sincerely,

Jian Sun, School of Athletic Training, Guangzhou Sport University, Guangzhou, P.R. China, E-mail: sunjian@gzsport.edu.cn . Tel:+8613728059899

---

## [Decision Letter · Decision Letter 3]

24 Oct 2022

PONE-D-22-02783R3Flywheel eccentric overload exercises versus barbell half squats for basketball players: which is better for induction of post-activation performance enhancement?PLOS ONE

Dear Dr. Sun,

Thank you for submitting your manuscript to PLOS ONE. After careful consideration, we feel that it has merit but does not fully meet PLOS ONE’s publication criteria as it currently stands. Therefore, we invite you to submit a revised version of the manuscript that addresses the points raised during the review process.

Statistical considerations still persist and it is of the uptmost importance to properly take into account reviewer's comments.

We look forward to receiving your revised manuscript.

Kind regards,

Laurent Mourot

Section Editor

PLOS ONE

Reviewers' comments:

Reviewer's Responses to Questions

**Comments to the Author**

1. If the authors have adequately addressed your comments raised in a previous round of review and you feel that this manuscript is now acceptable for publication, you may indicate that here to bypass the “Comments to the Author” section, enter your conflict of interest statement in the “Confidential to Editor” section, and submit your "Accept" recommendation.

Reviewer #1: All comments have been addressed

Reviewer #3: (No Response)

2. Is the manuscript technically sound, and do the data support the conclusions?

Reviewer #1: (No Response)

Reviewer #3: Yes

3. Has the statistical analysis been performed appropriately and rigorously? 

Reviewer #1: (No Response)

Reviewer #3: Yes

4. Have the authors made all data underlying the findings in their manuscript fully available?

Reviewer #1: (No Response)

Reviewer #3: Yes

5. Is the manuscript presented in an intelligible fashion and written in standard English?

Reviewer #1: (No Response)

Reviewer #3: Yes

6. Review Comments to the Author

Reviewer #1: (No Response)

Reviewer #3: My previously recommended revisions have again mostly been addressed. I have one comment remaining which I believe represents a major assumption within the manuscript. The authors recognise this but it should be made clear throughout the article.

You have confirmed within your response that the statistical analysis treats both exercises as having equal low, medium, or high resistance. The results are therefore inherently based on the degree to which the intensities are well matched. You recognised this within your reviewer response, but I believe it should be made clear throughout all sections of the manuscript so that (potentially uninformed) readers can make their own interpretation of the results based on this assumption.

7. PLOS authors have the option to publish the peer review history of their article (what does this mean?). If published, this will include your full peer review and any attached files.

Reviewer #1: No

Reviewer #3: No

---

## [Author Response · Author response to Decision Letter 3]

26 Oct 2022

Dear editor and dear reviewers

Regarding the manuscript ID (NO.PONE-D-22-02783) entitled " Eccentric overload exercises versus loaded half squats for basketball players: which is better for induction of postactivation potentiation?". Thank you for your letter and the reviewers' comments on our manuscript entitled " Eccentric overload exercises versus loaded half squats for basketball players: which is better for induction of postactivation potentiation?" (NO.PONE-D-22-02783) were evaluated. These comments were valuable and helpful to us in revising and improving the paper, as well as important guidance for our research. We have carefully studied these comments and made revisions, which we hope will be approved by all of you. The revised parts are marked in red in the thesis. The main corrections in the paper and the responses to the reviewers are listed below.

Comments to the Author

1. If the authors have adequately addressed your comments raised in a previous round of review and you feel that this manuscript is now acceptable for publication, you may indicate that here to bypass the “Comments to the Author” section, enter your conflict of interest statement in the “Confidential to Editor” section, and submit your "Accept" recommendation.

Reviewer #1: All comments have been addressed

Reviewer #3: (No Response)

2. Is the manuscript technically sound, and do the data support the conclusions?

Reviewer #1: (No Response)

Reviewer #3: Yes

3. Has the statistical analysis been performed appropriately and rigorously?

Reviewer #1: (No Response)

Reviewer #3: Yes

4. Have the authors made all data underlying the findings in their manuscript fully available?

Reviewer #1: (No Response)

Reviewer #3: Yes

5. Is the manuscript presented in an intelligible fashion and written in standard English?

Reviewer #1: (No Response)

Reviewer #3: Yes

6. Review Comments to the Author

Reviewer #1: (No Response)

Reviewer #3: My previously recommended revisions have again mostly been addressed. I have one comment remaining which I believe represents a major assumption within the manuscript. The authors recognise this but it should be made clear throughout the article.

You have confirmed within your response that the statistical analysis treats both exercises as having equal low, medium, or high resistance. The results are therefore inherently based on the degree to which the intensities are well matched. You recognised this within your reviewer response, but I believe it should be made clear throughout all sections of the manuscript so that (potentially uninformed) readers can make their own interpretation of the results based on this assumption.

Responds: We agree with you so much that we have revised the text. Please see L138-140, L486-490.

---

## [Editor Report · Decision Letter 4]

27 Oct 2022

Flywheel eccentric overload exercises versus barbell half squats for basketball players: which is better for induction of post-activation performance enhancement?

PONE-D-22-02783R4

Dear Dr. Sun,

We’re pleased to inform you that your manuscript has been judged scientifically suitable for publication and will be formally accepted for publication once it meets all outstanding technical requirements.

Kind regards,

Laurent Mourot

Section Editor

PLOS ONE